# Condensin I subunit Cap-G is essential for proper gene expression during the maturation of post-mitotic neurons

Amira Hassan[1], Pablo Araguas Rodriguez[1], Stefan K Heidmann[2], Emma L Walmsley[1], Gabriel N Aughey[1]*, Tony D Southall[1]*

[1]Department of Life Sciences, Imperial College London, London, United Kingdom; [2]Lehrstuhl für Genetik, University of Bayreuth, Bayreuth, Germany

**Abstract** Condensin complexes are essential for mitotic chromosome assembly and segregation during cell divisions, however, little is known about their functions in post-mitotic cells. Here we report a role for the condensin I subunit Cap-G in *Drosophila* neurons. We show that, despite not requiring condensin for mitotic chromosome compaction, post-mitotic neurons express Cap-G. Knockdown of Cap-G specifically in neurons (from their birth onwards) results in developmental arrest, behavioural defects, and dramatic gene expression changes, including reduced expression of a subset of neuronal genes and aberrant expression of genes that are not normally expressed in the developing brain. Knockdown of Cap-G in mature neurons results in similar phenotypes but to a lesser degree. Furthermore, we see dynamic binding of Cap-G at distinct loci in progenitor cells and differentiated neurons. Therefore, Cap-G is essential for proper gene expression in neurons and plays an important role during the early stages of neuronal development.

**\*For correspondence:**
g.aughey@imperial.ac.uk (GNA);
t.southall@imperial.ac.uk (TDS)

**Competing interests:** The authors declare that no competing interests exist.

## Introduction

Differentiated neurons are post-mitotic cells – they lack the ability to further divide to produce daughter cells. However, newly born neurons are not immediately ready to synapse with other neurons, nor generate action potentials. These immature cells undergo morphological changes, generating dendrites and axons, which will eventually form synapses with target cell(s) (*Cajal, 1890*). Furthermore, newly born neurons display a level of developmental plasticity that is not apparent in terminally differentiated neurons, having been shown to have the potential to dedifferentiate under conditions that do not affect more mature cells (*Southall et al., 2014*; *Zacharioudaki et al., 2019*). However, underlying changes at the chromatin level are only just starting to be investigated. Changes in gene expression, chromatin accessibility and the 3D genome organisation all occur as neurons differentiate from progenitor cells (*Yuen and Gerton, 2018*; *Hirano, 2016*; *Ganji et al., 2018*; *Kschonsak et al., 2017*; *Terakawa et al., 2017*; *Oliveira et al., 2005*). Currently, little is known about the mechanisms underlying this transition and the molecular factors that coordinate it.

During mitosis eukaryotic DNA is packaged into condensed chromosomes for efficient segregation between daughter cells. The formation of these highly organised chromosome structures is achieved largely by the actions of the condensin complex (*Yuen and Gerton, 2018*). Components of the condensin complex form a loop through which DNA is constrained under the regulation of ATP-dependent SMC subunits in a similar manner to the related cohesin complex (*Yuen and Gerton, 2018*). Most eukaryotes are thought to have two condensin complexes - condensin I and II. Both condensin complexes share a heterodimer of SMC subunits (SMC2/4) and differ based on inclusion of paralogous kleisin subunits (Cap-H/H2), and two HEAT-repeat subunits (CAP-G/G2 and CAP-D2/D3) (*Hirano, 2016*; *Figure 1A*). Each condensin subunit plays a crucial role in packaging long strands of DNA into chromosomes via asymmetric loop extrusion (*Ganji et al., 2018*; *Kschonsak et al.,*

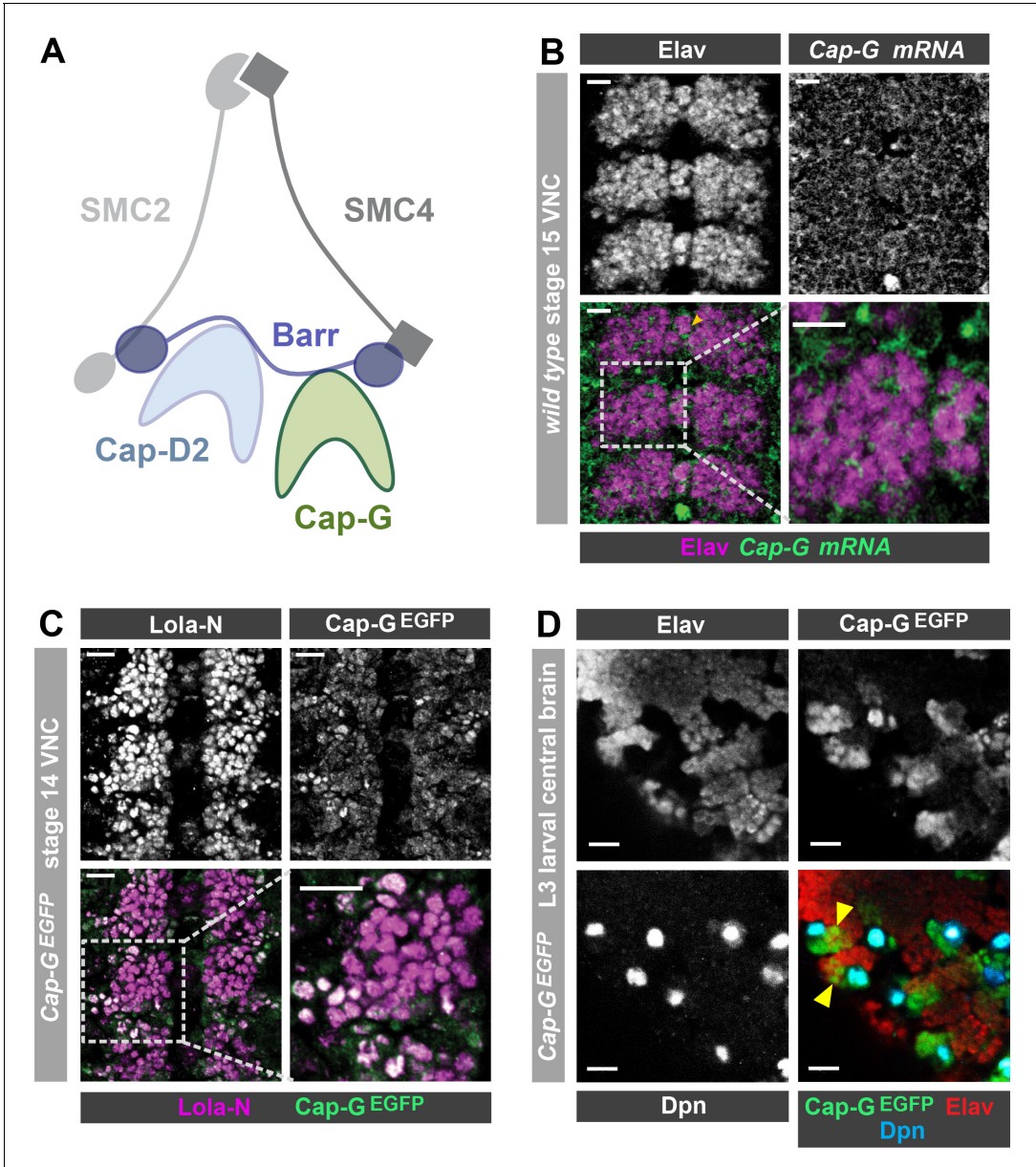

**Figure 1.** Cap-G expression in *Drosophila* neurons. (**A**) Schematic representation of the *Drosophila* condensin I complex. (**B**) *w^1118* embryo (stage 15, anterior top). *Cap-G* mRNA is ubiquitously present in neuronal cytoplasm, neurons marked by Elav. (**C**) *Cap-G^EGFP* embryo (stage 14, anterior top). Cap-G^EGFP co-localises with Lola-N in neuronal nuclei of the VNC. (**D**) Optic lobe of 3^rd instar larvae *Cap-G^EGFP*. Cap-G is strongly present in NSCs marked by Dpn. Cap-G is present in neurons (Elav positive) in proximity of NSCs (yellow arrowheads). Scale bars 10 μm.

The online version of this article includes the following figure supplement(s) for figure 1:

**Figure supplement 1.** Cap-G^EGFP expression in neurons.

2017; *Terakawa et al., 2017*) and ensuring proper individualisation of the chromosomes during cell divisions (*Oliveira et al., 2005*).

Several gene silencing mechanisms have also been linked to condensin activity. In yeast, chromatin compaction, driven by condensin, represses transcription in quiescent cells (*Swygert et al., 2019*). This is supported by observations in mouse T-cells, where condensin II depletion causes chromatin decompaction and an increase in gene expression, disrupting cellular quiescence (*Rawlings et al., 2011*). In *Drosophila*, condensin I is implicated in regulation of position effect variegation (PEV) (*Dej et al., 2004*; *Savvidou et al., 2005*; *Cobbe et al., 2006*). Heterozygous *Cap-G*

mutants show wing notches and rough eye phenotypes in flies, which is attributed to a regulatory role of Cap-G in heterochromatin gene expression (*Dej et al., 2004*). Furthermore, the *Drosophila* Cap-H orthologue, Barren, interacts with the chromatin-repressing Polycomb complex to silence homeotic genes. In *C. elegans*, the dosage compensation complex (DCC), closely related to condensin I, binds to DNA inducing transcriptional repression (*Meyer, 2010*). Moreover, depletion of condensin II shows an upregulation in gene expression due to disrupted gene silencing (*Kranz et al., 2013*). In murine cells, Cap-G2 has a potential role in erythroid cell differentiation by repressing transcription via chromatin condensation (*Xu et al., 2006*). However, some experiments in yeast argue that condensin does not directly regulate gene expression. For example, a recent study demonstrated that condensin depletion results in genome decompaction but has no effect on overall gene expression (*Paul et al., 2018*).

These studies observe the roles of condensin in mitotically proliferating cells, but there is limited knowledge of cell-specific, post-mitotic roles for condensin. Condensin mutations in mitotic and interphase cells ultimately disrupt post-mitotic daughter cells, leading to severe phenotypes. For example, in mouse neuronal stem cells (NSCs), condensin II mutations disrupted nuclear architecture and resulted in apoptosis of NSCs and post-mitotic neurons (*Nishide and Hirano, 2014*). One recent study showed that RNA-levels in *S. cerevisiae* are disrupted upon condensin inactivation due to the well-known phenotype of chromosomal mis-segregation during anaphase (*Hocquet et al., 2018*). This study points towards condensin having no direct role on transcription and raises the possibility that previous studies implicating condensin in gene expression may be suffering from artefacts resulting from aberrant chromosome segregation. Furthermore, condensin inactivation in differentiated mouse hepatocytes showed no changes in chromatin folding or gene expression (*Abdennur, 2018*). Conversely, a post-mitotic role for condensin II has been demonstrated in *Drosophila*, in which Cap-D3 regulates transcriptional activation of anti-microbial gene clusters in fat body cells (*Longworth et al., 2012*).

Despite the wealth of knowledge on condensin complexes, their role in post-mitotic cells remains unclear. In this study we reveal a novel role for the condensin I subunit Cap-G in *Drosophila* post-mitotic neurons. We observed Cap-G expression and localisation in the *Drosophila* central nervous system (CNS) in vivo. Cell-specific knockdown of Cap-G in neurons resulted in severe developmental arrest, behavioural defects, and an overall disruption of gene expression in the CNS. Knockdown animals exhibit a downregulation of neuron-specific genes and an ectopic upregulation of non-CNS-specific genes. Finally, Cap-G DNA binding profiles dynamically change between neuronal stem cells (NSCs) and post-mitotic neurons. The discovery of a neuronal role for Cap-G highlights the importance of studying condensin proteins in a post-mitotic context, to better understand their role in the regulation of gene expression.

## Results

### Cap-G is present in post-mitotic neurons

Upon conducting a yeast-2-hybrid screen to look for proteins interacting with the neuron-specific transcription factor Lola-N, we were surprised to identify the condensin complex component Cap-G as a potential interacting partner. Cap-G is a HEAT-repeat containing subunit exclusive to condensin I (*Herzog et al., 2013*; *Figure 1A*). Whilst condensin activity has been characterised in neural stem cells (*Nishide and Hirano, 2014*), the role of condensin complexes in post-mitotic neurons has not yet been studied in any species. Published RNA-seq data indicates significant levels of condensin subunit transcripts in neurons, including *Cap-G* (*Leader et al., 2018*; *Berger et al., 2012*). Therefore, we decided to investigate whether Cap-G was indeed present in *Drosophila* post-mitotic neurons. To characterise *Cap-G* expression in the central nervous system (CNS), we carried out fluorescent in-situ hybridisation for *Cap-G* mRNA on $w^{1118}$ embryos (stage 15). We observed ubiquitous expression of *Cap-G* in all cells of the Ventral Nerve Cord (VNC) of *Drosophila* embryos (*Figure 1B*). *Cap-G* mRNA was clearly detectable in neurons marked by the pan-neuronal marker Elav, indicating post-mitotic expression of *Cap-G* in the CNS. We also observe broad expression of *Cap-G* in the larval central nervous system in which *Cap-G* mRNA was detectable at similar levels in the neural stem cells (NSCs) and neurons (*Figure 1—figure supplement 1A*).

To investigate the distribution of Cap-G protein in the *Drosophila* CNS we utilised a *Cap-G*<sup>EGFP</sup> CRISPR knock-in line (*Kleinschnitz, 2020*). This line expresses C-terminally EGFP-tagged Cap-G from its endogenous locus, and therefore recapitulates native Cap-G expression. These flies are homozygous viable and fertile and display no apparent ill effects as a result of the EGFP tag. In stage 14 embryos we observed strong Cap-G<sup>EGFP</sup> signal in NSCs (*Figure 2A*) and we were also able to detect nuclear EGFP in post-mitotic neurons, which colocalised with the neuronal transcription factor Lola-N (*Southall et al., 2014*; *Figure 1C*). As expected, strong GFP signal was observed in mitotic cells in the larval CNS, particularly in NSCs and the optical proliferation centre (*Figure 1D*,

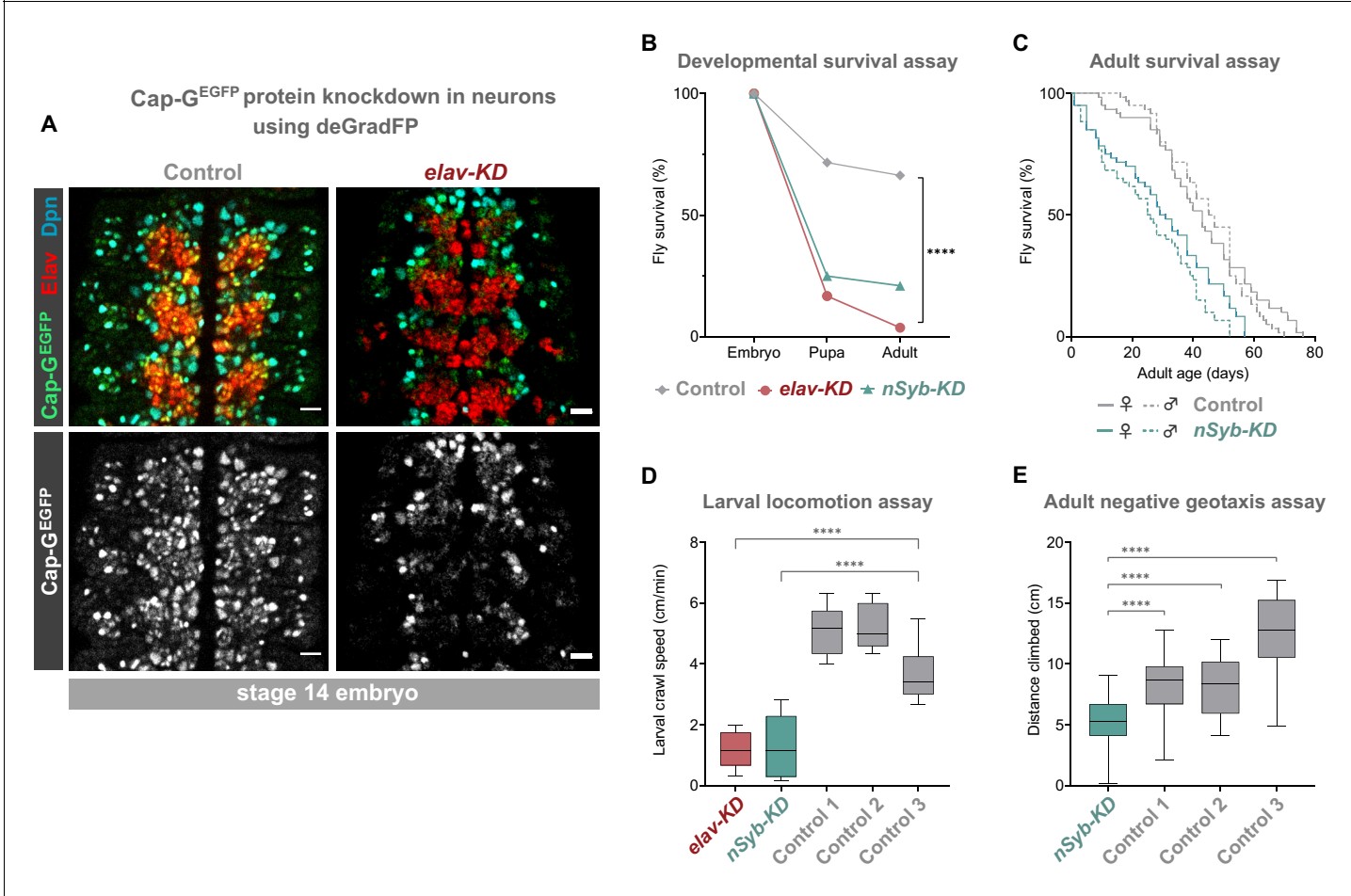

**Figure 2.** Cap-G knockdown in neurons results in premature developmental arrest and severe mobility defects. (**A**) Effective knockdown of Cap-G<sup>EGFP</sup> in neurons. Embryonic VNC (stage 14, anterior top) showing reduced Cap-G<sup>EGFP</sup> levels in neurons marked by Elav in *elav-KD* compared to *Cap-G*<sup>EGFP</sup> control. Cap-G<sup>EGFP</sup> is still detected in NSCs marked by Dpn in both *Cap-G*<sup>EGFP</sup> control and *elav-KD*. (**B**) Survival of developing flies recorded at pupal and adult stages. *Cap-G*<sup>EGFP</sup> flies were used as the control genotype. Survival expressed as a percentage of the initial total number of embryos. 300 biological replicates/genotype. Logrank test and weighted Gehan-Breslow-Wilcoxon model (****, p<0.0001). (**C**) Survival of adult *nSyb*-KD and *Cap-G*<sup>EGFP</sup> control recorded daily until all flies deceased. Survival expressed as a percentage of starting flies, 60 biological replicates/genotype. Logrank test and weighted Gehan-Breslow-Wilcoxon model (****, p<0.0001). (**D**) Locomotion assay of L3 larvae from three independent experiments. Control 1 = *Cap G*<sup>EGFP</sup>; *UAS-deGradFP*. Control 2 = *elav-GAL4*; *Cap-G*<sup>EGFP</sup>. Control 3 = *Cap G*<sup>EGFP</sup>; *nSyb-GAL4*. Mean crawling speed from 10 technical replicates for each of the 30 biological replicates/genotype. Kruskal-Wallis test one-way ANOVA (****, p<0.0001). (**E**) Negative geotaxis assay from three independent experiments. Control 1 = *Cap G*<sup>EGFP</sup>; *nSyb-GAL4*. Control 2 = *Cap G*<sup>EGFP</sup>; *UAS-deGradFP*. Control 3 = *nSyb-GAL4*; *UAS-deGradFP*. Mean distance climbed from 10 technical replicates for each of the 30 biological replicates/genotype. Kruskal-Wallis test one-way ANOVA (****, p<0.0001).

The online version of this article includes the following figure supplement(s) for figure 2:

**Figure supplement 1.** Characterisation of Cap-G knockdown with *elav-GAL4*.
**Figure supplement 2.** Characterisation of Cap-G knockdown with *elav-GAL4*.
**Figure supplement 3.** Apoptosis in *elav-KD*.

*Figure 1—figure supplement 1B*). Cap-G not only co-localised with NSC-marker Deadpan (Dpn), but also with neuronal marker Elav in the larval central brain (*Figure 1D*). Interestingly, EGFP signal was strongest in Elav-positive neurons in closest proximity to ganglion mother cells (GMCs) and NSCs in both embryos and larvae, indicating that Cap-G may play a more prominent role in newly born neurons.

To determine whether Cap-G was the only member of the condensin I complex to be present in post-mitotic neurons, we also examined *barren*[EGFP] animals which express a C-terminally EGFP-tagged Barren variant from *the barren* genomic locus. Barren[EGFP] was clearly detected in all cells of the embryonic VNC as we saw with Cap-G (*Figure 1—figure supplement 1C*). Interestingly, we observed that Barren appeared to be predominantly in the cytoplasm of neurons, with much lower signal apparent in the nucleus. This result is in agreement with previous reports which have shown that Barren is primarily localised in the cytoplasm of interphase cells (in which Cap-G appears largely nuclear) (*Herzog et al., 2013*; *Oliveira et al., 2007*). Since Cap-G showed the most distinct nuclear localisation, we decided to focus on the neuronal role of Cap-G for the rest of this study. Overall, these data confirm that condensin expression is prevalent in the post-mitotic cells of the fly CNS and is maintained throughout development, from embryonic to larval stages, indicating that condensin I may have a previously unappreciated role in post-mitotic neurons.

## The deGradFP system produces robust neuron specific knockdown of Cap-G

The presence of Cap-G in neurons suggests a post-mitotic function for condensin in the CNS. To investigate this putative neuronal role, we used the deGradFP system to create neuron-specific Cap-G knockdown (KD) animals (*Caussinus et al., 2012*). When combined with *GAL4/UAS*, this system allows for highly specific knockdown of target proteins in a cell-type of interest. By combining deGradFP with *Cap-G*[EGFP] flies, we were able to successfully induce degradation of Cap-G. As expected, ubiquitous degradation of Cap-G-EGFP with *tubulin-GAL4* resulted in 100% embryonic lethality. This is consistent with the essential role of condensin proteins during mitosis as well as previous reports of *Cap-G* mutant embryonic lethality in *Drosophila* (*Dej et al., 2004*; *Jäger et al., 2005*).

We induced Cap-G knockdown in neurons, using two post-mitotic drivers, *elav-GAL4* and *nSyb-GAL4. elav-GAL4* is expressed in all neurons, from newly born to mature, whilst *nSyb-GAL4* drives expression solely in more mature neurons in which synapse formation has begun. We performed knockdown experiments using both *GAL4* lines to better characterise the role of Cap-G during neuronal maturation since we observed higher levels of Cap-G in newly differentiated neurons (*Figure 1C,D*). Cap-G knockdown experiments with these two drivers are referred to as *elav-KD* and *nSyb-KD* respectively hereafter. Immunostaining of *elav-KD* embryos showed a reduction in GFP signal in neurons but not in NSCs, indicating that a robust neuronal knockdown was achieved using the deGradFP method (*Figure 2A*). Previous reports have indicated that *elav-GAL4* can drive low level expression in embryonic NSCs (but not post-embryonic NSCs) (*Berger et al., 2007*). Since a recent study raised concerns that condensin knockdown phenotypes seen in interphase may in fact be due to defects in mitotic chromosome segregation (*Hocquet et al., 2018*) we undertook a careful characterisation of the *elav-KD* on NSCs.

To determine whether we could see any reduction in Cap-G[EGFP] levels in NSCs or GMCs, we quantified fluorescence levels in live mitotic cells of the VNC in *elav-KD* and *Cap-G*[EGFP] control embryos. We observed no difference in Cap-G[EGFP] fluorescence in mitotic cells between the control and *elav-KD* embryos (*Figure 2—figure supplement 1A*). We further analysed Cap-G[EGFP] signal intensity in fixed tissue using the NSC marker Deadpan (Dpn) as a counterstain. Quantification of Cap-G[EGFP] in Dpn positive cells revealed no significant difference in signal intensity (*Figure 2—figure supplement 1B,C*). Therefore we conclude that any low-level expression of deGradFP in NSCs is not sufficient to reduce Cap-G[EGFP] levels by a detectable amount. Due to the well characterised function of condensin during mitosis, we reasoned that degradation of Cap-G in NSCs would affect their ability to divide. Therefore, we quantified the total number of NSCs (marked by Dpn), the total number of dividing cells (marked by the M-phase marker pH3 *Giet and Glover, 2001*) and finally the number of actively dividing NSCs (Dpn and pH3-positive cells). We saw no differences in the overall number of NSCs (*Figure 2—figure supplement 2A*), dividing cells (*Figure 2—figure*

supplement 2B), or number of mitotic NSCs between control and *elav-KD* embryos (*Figure 2—figure supplement 2C,D*).

Depletion of condensin in the mouse cortex resulted in NSC and neuronal apoptosis (*Nishide and Hirano, 2014*). Similarly, significant apoptosis was seen in the proliferating cells of zebrafish retinas in Cap-G mutants (*Seipold et al., 2009*). Therefore, we sought to determine whether an increased amount of cell death was observable in *elav-KD* NSCs, which could be a result of premature Cap-G depletion. TUNEL staining of embryonic VNCs showed no significant difference in the number of apoptotic cells between *Cap-G*$^{EGFP}$ controls and *elav-KD* in non-neuronal cells (*Figure 2—figure supplement 3A–C*). Together, these data show that *elav-KD* does not significantly lower Cap-G levels in NSCs, it does not affect NSC numbers, nor the number of dividing cells, and does not induce apoptosis. Therefore, we conclude that the *elav-KD* has no effect on NSCs and dividing cells at the embryonic stage, suggesting that with this method we are able to achieve a neuron specific, post-mitotic Cap-G knockdown. Although we did not see increased levels of apoptosis in progenitor cells, we did observe a small but statistically significant increase in the number of Elav positive apoptotic neurons, suggesting that degradation of Cap-G may result in decreased cell survival in post-mitotic neurons (*Figure 2—figure supplement 3C*).

## Neuron-specific knockdown of Cap-G leads to behavioural phenotypes and reduced survival across development

To determine the impact of neuronal depletion of Cap-G on animal survival, we assayed the numbers of *elav-KD* and *nSyb-KD* flies pupating or eclosing as adult flies. We observed a severe developmental lethality phenotype in both neuronal Cap-G KD flies when compared to control *Cap-G*$^{EGFP}$ flies. In *elav-KD* animals, only 17% of embryos developed into pupae, and only 4% successfully eclosed to produce adult animals, compared to 71% and 66% reaching pupal or adult stages respectively in controls (*Figure 2B*). Embryos from *nSyb-KD* flies also displayed a severe survival defect, with only 25% pupating and 21% adults eclosing. Survival analysis using both the Logrank and the Gehan-Breslow-Wilcoxon tests showed a significant difference between the Cap-G$^{EGFP}$ KDs and control flies (p<0.0001, n = 300). Whilst the number of animals surviving to pupation was not significantly different between *elav-KD* and *nSyb-KD,* significantly more *nSyb*-KD flies survived to adult stages. The surviving 4% *elav-KD* adults all died shortly after eclosing.

Despite appearing morphologically normal, we investigated whether surviving *nSyb-KD* flies had any detectable neuronal dysfunction in adulthood. Survival assay for *nSyb-KD* flies revealed a significantly higher mortality rate than controls (*Figure 2C*). Survival analysis using both the Logrank and the Gehan-Breslow-Wilcoxon tests showed a significant difference between the *nSyb-KD* and control flies (p<0.0001, n = 60). Overall, these results suggest that Cap-G in post-mitotic neurons is necessary for normal development of the CNS and survival.

Further to premature death, we analysed effects of Cap-G KD on larval locomotion ability, commonly used as a proxy to reveal defects in neuronal development (*Lanson et al., 2011*). Both *elav-KD* and *nSyb-KD* animals displayed locomotion defects when compared to controls (*Figure 2D*). On average, the Cap-G knockdown larvae moved 4 cm/min less than the control animals (p<0.0001, n = 30). We observed that mobility impairment carried over onto adult flies (*Figure 2E*). *elav-KD* adults were unable to move after eclosion and died shortly after. A negative geotaxis (climbing) assay (*Gargano et al., 2005*) on *nSyb-KD* adults showed a compromised ability to climb, with the mean climbing distance of ~5 cm, as compared to ~9 cm or ~ 13 cm for control genotypes (Kruskal-Wallis test one-way ANOVA, p<0.0001, n = 30) (*Figure 2E*). It should be noted that since such a low proportion of flies survived to adult stages, these flies may have escaped the most severe consequences of Cap-G depletion.

## Neurons have increased levels of DNA damage due to post-mitotic Cap-G KD

We reasoned that due to the role of condensin complexes in organising chromosomes, depletion of Cap-G may have resulted in genomic instability, therefore we decided to assay DNA damage in Cap-G knockdown flies. Using a γ-H2Av antibody as a marker for DNA damage (*Lake et al., 2013*) and Elav as a neuronal marker, we quantified the percentage of γ-H2Av positive neurons in larval brains. In the larval CNS we observed an increase in neuronal DNA damage in both *elav-KD* and

nSyb-KD, when compared to *Cap-G*$^{EGFP}$ controls (*Figure 3A*). In *elav-KD* and *nSyb-KD* we observed a two/threefold increase of γ-H2Av+ neurons compared to controls, (Kruskal-Wallis test one-way ANOVA, p<0.001) (*Figure 3B*).

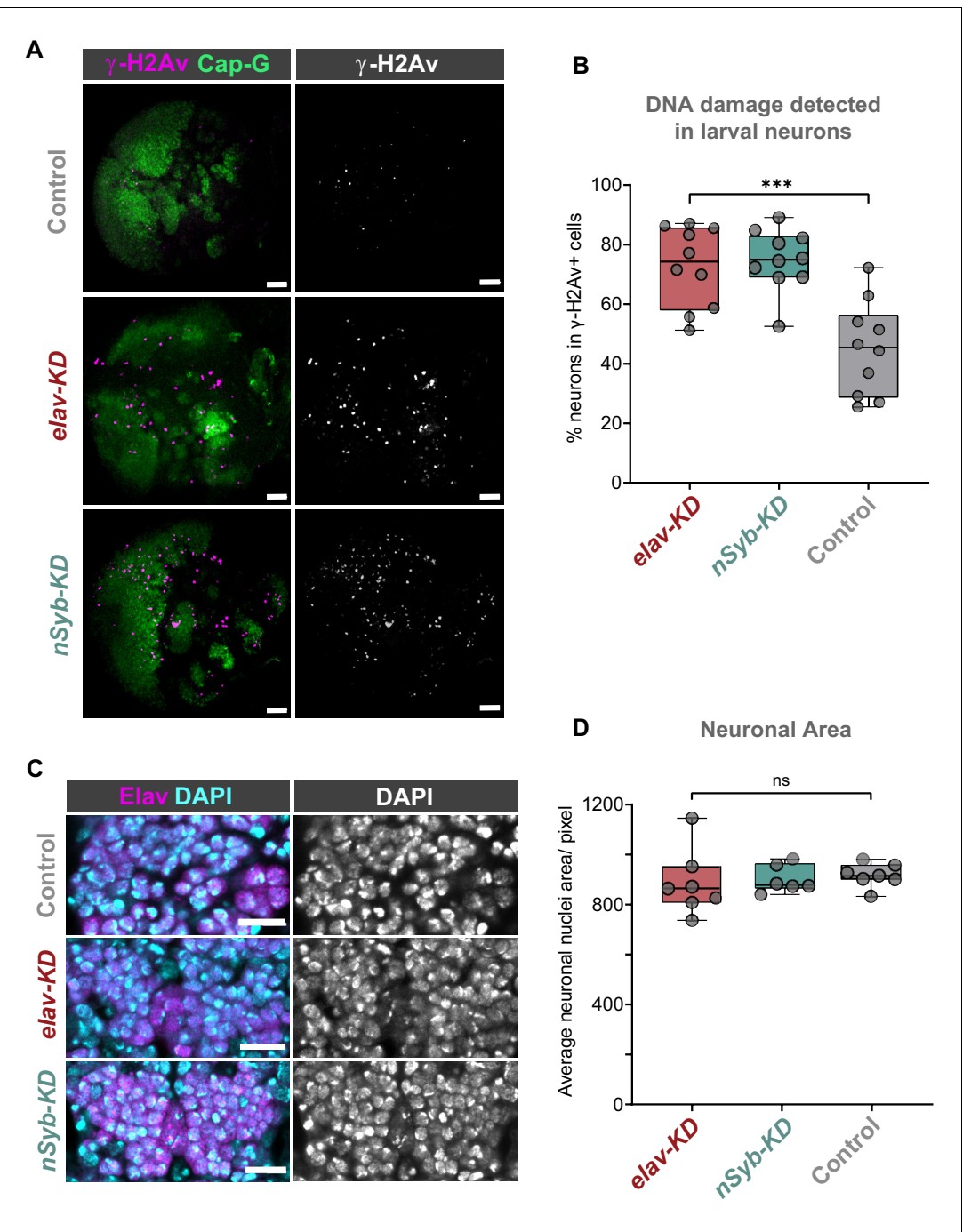

**Figure 3.** Cap-G knockdown leads to increased DNA damage in larval neurons. (**A**) Larval optic lobes for *Cap-G*$^{EGFP}$ controls and *Cap-G KD* genotypes. γ-H2Av indicates DNA damage, Cap-G signal is strongest in dividing cells. Scale bars 10 µm. (**B**) Quantification of γ-H2Av+ neurons as percentage of total γ-H2Av+ cells. 10 biological replicates per genotype. Kruskal-Wallis test one-way ANOVA (***, p<0.001). (**C**) Section of embryonic VNC (ventral view) stage 15 for all genotypes. Elav to mark neuronal nuclei and DAPI as nuclei stain. Control (*Cap-G*$^{EGFP}$), *elav-KD* and *nSyb-KD* genotypes. (**D**) Quantification of neuronal nuclei area. Pixel area average of at least 1000 cells per replicate was analysed. seven biological replicates per genotype. No statistical difference determined by one-way ANOVA (p>0.5).

Given that condensin complexes are implicated in chromatin condensation, it seemed reasonable to question whether Cap-G has any role in regulating chromatin structure or distribution in neurons. To assess whether Cap-G KD influenced overall chromatin condensation we quantified nuclear area in embryonic neurons. Control and KD embryos were stained for the neuronal marker Elav as well as DAPI to highlight overall DNA content (*Figure 3C*). We observed no significant difference in the size of nuclear area between Cap-G KD and controls when Cap-G was depleted with either *elav* or *nSyb-GAL4* (*Figure 3D*).

## Dynamic Cap-G association with chromatin in mitotic NSCs and post-mitotic neurons

Since we have observed Cap-G in the nuclei of post-mitotic neurons, we sought to characterise its association with chromatin to further illuminate its role in gene regulation in the CNS. We used Targeted DamID (TaDa) (*Southall et al., 2013*; *Aughey et al., 2019*) to profile cell-type specific Cap-G binding in NSCs and post-mitotic, differentiated neurons. For differentiated neurons, we continued to use both *elav-GAL4* and *nSyb-GAL4* drivers to further characterise differences in Cap-G binding between newly born and fully differentiated neurons (*Figure 4A*). Cap-G binding in NSCs was determined using *wor-GAL4* so that we could compare the genomic localisation of Cap-G between actively dividing and post-mitotic cells. TaDa requires the expression of a fusion protein with the *E. coli* Dam methylase. Since we have seen that Cap-G remains functional when tagged with EGFP at the C-terminus, the fusion of Dam at the same site should not disrupt Cap-G localisation or function.

Two biological replicates were performed for each condition and significantly bound regions were identified using a custom peak calling method developed specifically for DamID data (see Materials and methods). The DamID methyl-PCR required for the TaDa experiments yielded strong amplification for all cell types, indicating that Cap-G-Dam associates with chromatin in post-mitotic cells. Sequencing revealed that Cap-G peaks are prevalent across the genome of all cell types. There is a strong positive correlation between biological replicates for each cell stage (Spearman's correlation, r = ~0.9) (*Figure 4—figure supplement 1A*). Moreover, principal component analysis shows that biological replicates per cell stage cluster together into three independent clusters, matching each cell stage analysed (*Figure 4—figure supplement 1B*).

Cap-G binding is highly variable between NSCs and differentiated neurons, suggesting specific roles in gene regulation in these cell types (*Figure 4A*, *Figure 4—figure supplement 1D*). Cap-G binds in the region of known NSC- specific genes such as *dpn* and *HLHmγ* (*Harding and White, 2018*; *Figure 4A*). The cell-cycle genes *CycE* and *stg* display a similar trend (*Figure 4A*, *Figure 4—figure supplement 1D*). Conversely, Cap-G peaks are detected at the neuronal gene *nSyb* in NSCs and *elav* neurons but are absent in nSyb neurons (*Figure 4A*). Moreover, Cap-G also binds to non-neuronal genes in all cell-types, such as *Unc-89*, which is expressed in muscle tissues, but not the CNS (*Gargano et al., 2005*; *Figure 4—figure supplement 1D*). Whilst binding of Cap-G is frequently seen in the gene body, in some cases peaks are observed in the surrounding locus as in the case of *stg* in which several peaks are present in the upstream regulatory region (*Figure 4—figure supplement 1D*). Overall, the number of genes bound by Cap-G is similar across cell types, but significant differences are apparent with uniquely bound genes detected in each cell-type (*Figure 4B*). Interestingly, marked differences are apparent in Cap-G binding between the two post-mitotic cell stages. Together these data suggest that Cap-G binding is cell-specific and dynamic across cell types, changing as the NSC differentiates and the neuron matures.

Cap-G peaks are strongly enriched in gene bodies and depleted in intergenic and non-coding regions (*Figure 4C*, *Figure 4—figure supplement 1C*). We did not observe significant binding of Cap-G at tRNA genes or rRNA genes in our data, as previously reported for condensin binding in *S. cerevisiae* (in which there is only one condensin complex) or condensin II binding in mouse ESCs (*D'Ambrosio et al., 2008*; *Yuen et al., 2017*).

## Chromatin accessibility is reduced at Cap-G binding sites across cell types

Previous evidence suggests that condensin associates with open chromatin and enhancer regions in *C. elegans* (*Kranz et al., 2013*). Similarly, the condensin II subunit, Cap-H2 localises at enhancer regions in *Drosophila* Kc167 cells (*Li et al., 2015a*). We asked whether Cap-G binding similarly

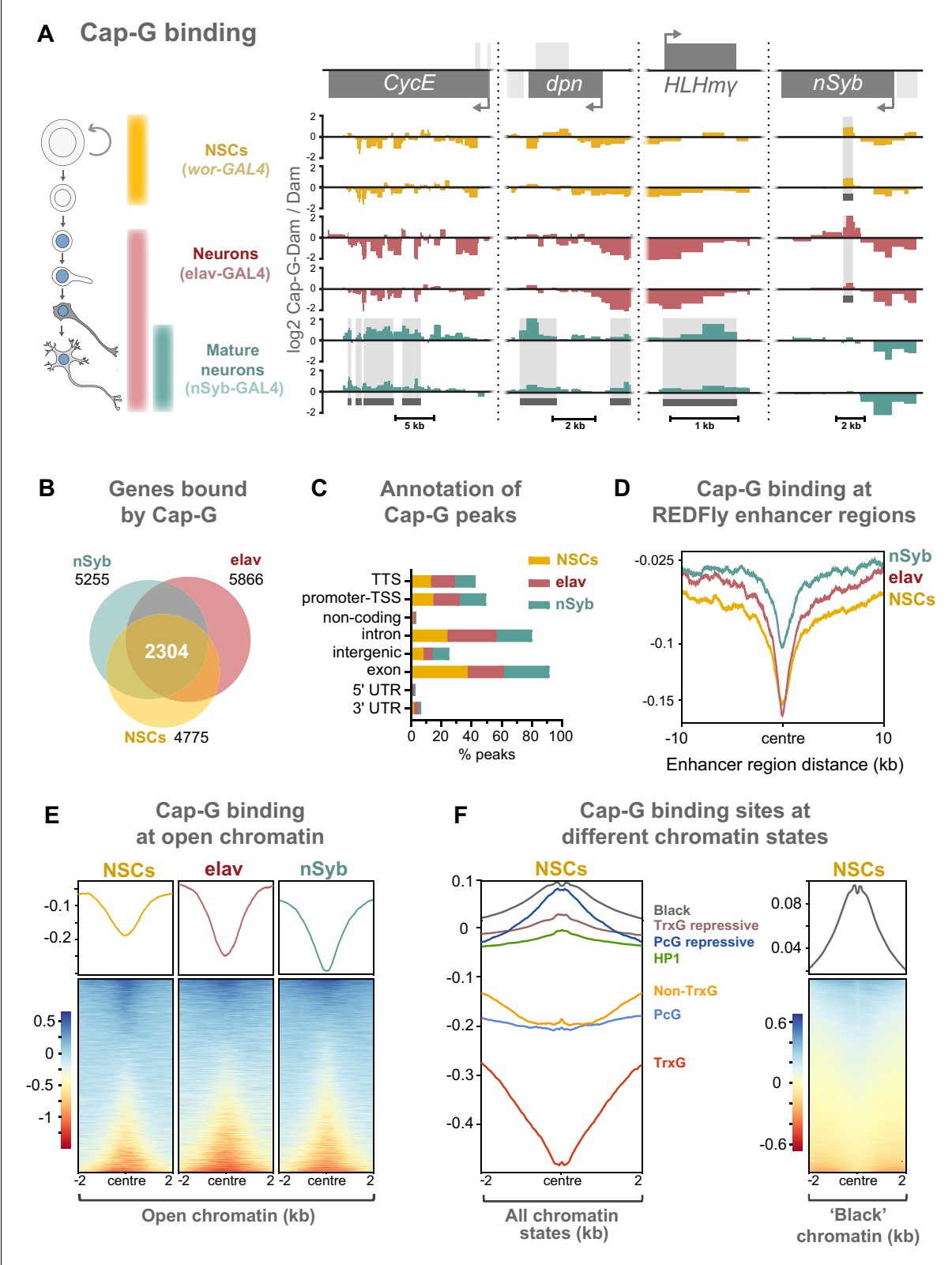

**Figure 4.** Cap-G binds to DNA in NSCs and neurons. (A) Cap-G binding at example loci in NSCs (*wor-GAL4*), all neurons including immature neurons (*elav-GAL4*) and mature (*nSyb-GAL4*) neurons. Light grey boxes on the gene annotation tracks represent other genes. Horizontal grey bars on the data tracks indicate statistically significant peaks. Y-axes display normalised ratio of $\log_2$ Cap-G-Dam/Dam. N.b. since TaDa data is normalised to Dam-only signal it is normal to observe 'negative peaks' at sites of depleted binding that have higher background Dam-only methylation. (B) Venn diagram
*Figure 4 continued on next page*

*Figure 4 continued*

showing unique genes bound by Cap-G and the total overlap between all cell types. (**C**) Genomic annotation of Cap-G peaks shows enrichment in gene bodies (introns and exons) whilst binding at non-coding regions are depleted. (**D**) Cap-G binding is depleted at known *Drosophila* Regulatory Elements (REDfly, http://redfly.ccr.buffalo.edu/index.php). Profiles plotted against centre of enhancer region and 10 Kb up/downstream. (**E**) Average Cap-G binding is depleted at accessible chromatin for all cell types. Plot shows 2 Kb up/downstream centre of an open chromatin region. Note that a subset of sites appear enriched for Cap-G binding (blue lines on heatmap) (**F**) Average Cap-G binding in different chromatin states in NSCs. Cap-G binding is enriched in repressive states (Black, HP1, TrxG-repressive, and PcG repressive), and depleted in permissive chromatin states (non-TrxG, TrxG, and PcG). Heatmap is shown for the most highly enriched 'black' state (non-HP1, non-PcG repressive state).

The online version of this article includes the following figure supplement(s) for figure 4:

**Figure supplement 1.** Analysis of Cap-G binding.
**Figure supplement 2.** Comparisons between chromatin accessibility and Cap-G binding.

occurs at known regulatory sequences in *Drosophila.* To address this we analysed Cap-G binding at all experimentally verified cis-regulatory modules (CRMs) from the REDfly database (*Rivera, 2018*; http://redfly.ccr.buffalo.edu/index.php) and observed a strong depletion of Cap-G binding at those sites, suggesting that Cap-G is not broadly associated with CRMs (*Figure 4D*).

Since many CRMs from this dataset may not be accessible in a given cell type, it was desirable to analyse Cap-G binding with respect to chromatin accessibility in the relevant cell-type of interest. To accomplish this we utilised the recently described Chromatin Accessibility TaDa method (CATaDa) (*Aughey et al., 2018*). This method takes advantage of the fact that the untethered Dam used to normalise our TaDa data preferentially methylates accessible chromatin, meaning that reanalysis of our control data can be used to determine open chromatin regions in the specific cell-types used. Furthermore, since these data are acquired from animals using the same drivers and treated under the same conditions as the experimental dam-fusions, it provides an ideal dataset for comparing open chromatin with Cap-G binding. Overall, open chromatin regions determined by CATaDa display a general depletion of Cap-G binding (*Figure 4E*). Whilst the majority of accessible sites were depleted for Cap-G binding, a subset of sites showed some enrichment for Cap-G. We also observed a negative correlation between chromatin accessibility and Cap-G binding across all cell types (*Figure 4—figure supplement 2A,B*).

It has been demonstrated that chromatin can be divided into several discrete states depending on the occupancy of various key proteins (*Filion et al., 2010*). These states include repressive and permissive chromatin environments. Recently published data allowed us to compare Cap-G binding to chromatin states in NSCs (*Marshall and Brand, 2017*). We find that Cap-G binding is most strongly enriched in repressive chromatin states (*Figure 4F*). Cap-G was particularly strongly associated with 'black' chromatin – a prevalent repressive state that does not incorporate traditional heterochromatin markers. In contrast, Cap-G binding was relatively depleted at permissive chromatin states, particularly the 'red' TrxG state (*Figure 4F*). Overall, our results suggest that accessible chromatin is depleted at Cap-G binding sites indicating that condensin I does not bind to known enhancer and regulatory regions in *Drosophila* NSCs and neurons. Together these data suggest that Cap-G binding of chromatin in neurons may have a regulatory role independent of binding to accessible chromatin and may promote, or be recruited to, repressive chromatin environments.

## Knockdown of Cap-G in neurons leads to misregulation of gene expression in the CNS

As Cap-G is expressed in neurons and demonstrates specific DNA binding properties in differentiated cells, the role of Cap-G in neuronal gene regulation was investigated. We performed RNA-seq experiments on the CNS of neuron-specific Cap-G knockdown flies. Due to the differences we had previously observed between *elav-KD* and *nSyb-KD* phenotypes, we continued to use both drivers to investigate differences in gene expression when Cap-G is depleted either in newly born and mature neurons (*elav-KD*), or more mature neurons (*nSyb-KD*). Cap-G knockdown resulted in disruption of the neuronal transcriptome, with a significant number of genes differentially expressed using both *elav* and *nSyb* drivers (*Figure 5A*, *Figure 5—figure supplement 1A,B*). *elav-KD* resulted in 1360 upregulated and 1308 downregulated genes, whilst in the *nSyb-KD* we observed a more modest effect on gene expression, with 152 upregulated and 126 downregulated genes. Of genes that

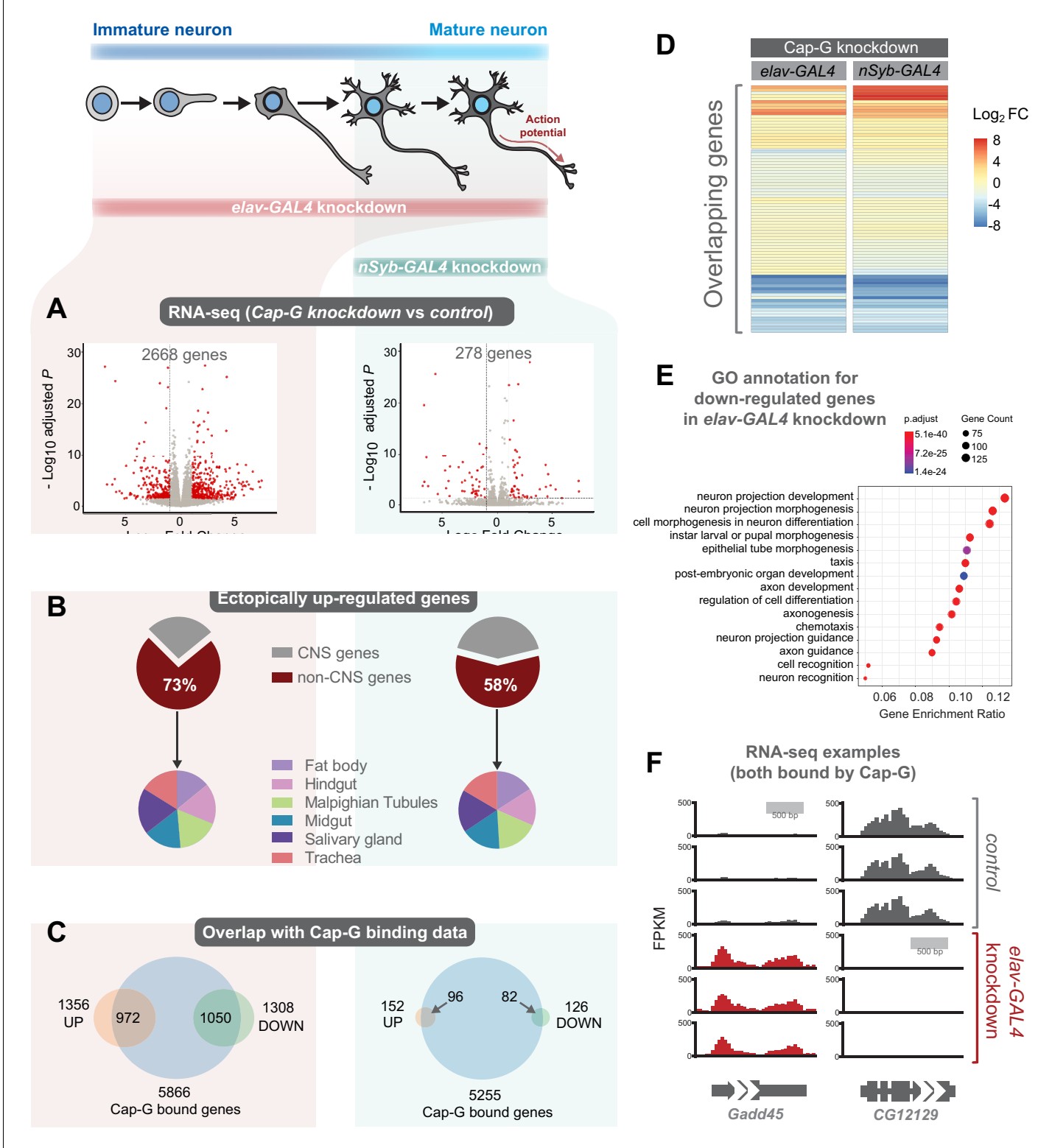

**Figure 5.** Cap-G knockdown in neurons display severe misregulation of gene expression. (**A**) Volcano plots showing differentially expressed genes in *elav-KD* and *nSyb-KD* respectively. Significant differential expression marked in red by a Log2 fold change > 1 or <-1 and FDR < 0.05. Control genotypes are *elav-GAL4; UAS-deGradFP* and *nSyb-GAL4; UAS-deGradFP* for *elav-KD* and *nSyb-KD* respectively. (**B**) Pie charts showing tissue of origin expression data for upregulated genes in Cap-G KD datasets taken from FlyAtlas. The majority of genes are not normally expressed in the CNS but in alternative tissues. (**C**) Venn diagrams showing significant overlap between Up/Down regulated genes and Cap-G DamID peaks for elav and nSyb

*Figure 5 continued on next page*

*Figure 5 continued*

neurons respectively, Fisher's exact test (p<1×10$^{-20}$). (D) Heatmap of overlapping differentially expressed genes between *elav-KD* and *nSyb-KD*. (E) Enriched Gene Ontology analysis of downregulated genes from *elav-KD* shows mostly neuron-specific terms. (F) Example of differentially expressed genes directly bound by Cap-G in *elav-KD* (all three replicates displayed).

The online version of this article includes the following figure supplement(s) for figure 5:

**Figure supplement 1.** Analysis of RNA-seq data.

are differentially expressed in both experiments (83 genes), a similar pattern of up/downregulation was observed (*Figure 5D*).

Analysis of Enriched Gene Ontology (GO) terms shows that downregulated genes in *elav-KD* were tissue-specific, with neuron-specific terms such as 'neuron projection development' and 'cell morphogenesis in neuron differentiation' being highly enriched (*Figure 5E*). Gene expression data extracted from FlyAtlas 2 (*Leader et al., 2018*) revealed that the majority of upregulated genes are not normally expressed in the larval CNS (*Figure 5B*). 73% of upregulated genes in *elav-KD*, and 58% in *nSyb-KD* are non-CNS specific, belonging to tissues such as the Midgut, Hindgut and Fat body. Interestingly, the tissue distribution of these ectopically expressed genes is very similar in both knockdown scenarios. This is further confirmed by GO term analysis of upregulated genes that yielded a variety of non-CNS specific GO terms for both knockdowns, such as 'nucleotide metabolic process' and 'midgut development' (*Figure 5—figure supplement 1C,D*). These data suggest that Cap-G contributes to the repression of non-neuronal gene expression following terminal differentiation.

To determine whether misregulation of gene expression was a direct consequence of loss of Cap-G binding in the genome we compared our RNA-seq data to the previously described Cap-G binding profiles (*Figure 4*). We observed a significant overlap between Cap-G bound genes in our TaDa data and the differentially expressed genes for both *elav* and *nSyb-KD* (*Figure 5C,F*). In *elav* expressing neurons, 80% of downregulated genes and 72% of upregulated genes were directly bound by Cap-G. These overlaps are highly significant (Fisher's exact test, p<10$^{-20}$). In mature neurons, 63% of upregulated genes and 65% of downregulated genes were bound by Cap-G (Fisher's exact test, p<10$^{-20}$). These data suggest that the majority of misregulated gene expression observed may be a result of direct binding by Cap-G.

## Discussion

Whilst condensin function in chromosome segregation is well defined, non-canonical roles for condensin complexes in the regulation of gene expression are less well understood. Despite this, recent studies have provided evidence of condensin proteins contributing to gene regulation and chromatin organisation (*Lau and Csankovszki, 2014*). Whilst it seems certain that condensin complexes play a role in gene regulation, some of these studies may suffer from technical artefacts relating to premature condensin knockdown in dividing cells (*Hocquet et al., 2018*). Furthermore, studies examining condensin function outside of mitosis have drawn conflicting conclusions (*Abdennur, 2018*; *Longworth et al., 2012*). In this manuscript we aimed to characterise the role of a condensin I protein (Cap-G) in a previously unstudied post-mitotic cell type. We found that Cap-G plays a significant part in regulating gene expression in neurons which is necessary for normal nervous system function.

### A post-mitotic role for Cap-G/condensin I in neurons

We observed nuclear localisation of Cap-G in *Drosophila* NSCs and neurons during development. The presence of *Drosophila* Cap-G in the nuclei of post-mitotic neurons mirrors the reported nuclear localisation of Cap-G in interphase (*Herzog et al., 2013*). Previous analyses of condensins in the central nervous system showed that condensin I and II are both involved in murine NSC division, but 'largely absent' from neurons (*Nishide and Hirano, 2014*). It is possible that the relative intensity of signal observed from dividing and post-mitotic cells masked the presence of condensin subunits in this instance. However, both our in situ hybridisation and fluorescently tagged protein imaging, as well as previously published RNA-seq data (*Leader et al., 2018*), strongly suggest that Cap-G is

prevalent in *Drosophila* post-mitotic neurons. Further studies will be required to determine whether this property is conserved in other species.

Our results show that Cap-G depletion leads to alterations in gene expression in both early and mature neurons in *Drosophila* larvae. We observed that neuron-specific genes are downregulated, whilst non-CNS genes are ectopically upregulated in neurons depleted of Cap-G, suggesting that Cap-G activity contributes to maintenance of the neuronal transcriptome. These observations are supported by previous studies in interphase and mitotic cells which show that Condensin I and II regulate cell-specific gene expression in multiple species (*Yuen et al., 2017*; *Rawlings et al., 2011*; *Li et al., 2015b*). However, only a single study reports a post-mitotic role for condensin, specifically condensin II subunit Cap-D3, in *Drosophila* (*Longworth et al., 2012*). The authors describe Cap-D3 regulating cell-specific gene expression in fat body cells, where it binds to and transcriptionally activates innate-immunity genes, including antimicrobial peptides. Interestingly, adult flies depleted of Cap-D3 have impaired immune response and fail to effectively clear bacteria (*Longworth et al., 2012*). Therefore, condensin I and II subunits may have different roles according to developmental stages, cell types, and complex incorporation.

Our data suggest that DNA damage is significantly increased in Cap-G depleted neurons. Previous studies in yeast have implicated the condensin complex in the regulation of the DNA damage response during interphase (*Aono et al., 2002*; *Chen et al., 2004*). Moreover, *Drosophila* condensin II has been implicated in inhibiting double stranded break (DSB) formation (*Schuster et al., 2013*). In human cells condensin I was shown to have a function in single stranded break (SSB) specific DNA damage repair but was not involved in DSB repair (*Heale et al., 2006*). In contrast, the antibody used in our DNA damage experiments specifically recognises the response to DSBs. So far, Condensin I has not been shown to function in DNA damage pathways in *Drosophila*, therefore it is possible that condensin I is involved in DSB repair in flies, in contrast to mammalian cells. However, Cap-G/condensin I may be required to maintain genomic stability in neurons independently of the DNA damage response. It is unclear whether the DNA damage we observe is responsible for the changes in gene expression. Given that we see similar levels of DNA damage in neurons of both *elav* and *nSyb* mediated knockdown, which have significantly different levels of aberrant gene expression, it is likely that increased DNA damage is not directly responsible for the changes we see in gene expression.

## Cap-G mediated regulation of gene expression

To date, the role of condensin in transcription regulation has only been speculated for interphase and dividing cells. For example, condensin involvement in transcriptional repression has been suggested in *C. elegans* (*Kranz et al., 2013*), mouse T-cells (*Rawlings et al., 2011*), and yeast (*Swygert et al., 2019*). However, there is conflicting evidence in the literature on the role of condensin in gene expression. A recent study in fission and budding yeast suggests that condensin has no direct effect on gene expression and that changes to RNA levels occur as an indirect effect of chromosome mis-segregation in condensin-depleted cells (*Hocquet et al., 2018*). Moreover, recent studies in yeast and mouse hepatocytes showed no significant changes in gene expression upon condensin depletion (*Paul et al., 2018*; *Abdennur, 2018*). Our data is exclusively post-mitotic therefore there is no effect on chromosome stability, emphasising that the effects we see on neuronal gene expression are a direct result of Cap-G depletion. Furthermore, a large proportion of the affected loci appear to be associated with Cap-G, suggesting that Cap-G could have a direct transcriptional regulatory effect on its binding regions. This is also observed in *C. elegans*, where the dosage compensation complex binds to specific DNA regions and promotes transcriptional repression (*Meyer, 2010*; *McDonel et al., 2006*).

We demonstrated that Cap-G binds dynamically to DNA in NSCs and post-mitotic neurons. Each cell type displayed unique genes bound by Cap-G. Interestingly, Cap-G peaks are enriched mid-gene rather than at promoter regions as reported in *C. elegans* and chicken DT40 cells (*Kranz et al., 2013*; *Kim et al., 2013*). Moreover, we did not observe any overlap between Cap-G binding and TFIIIC targets, as previously described in multiple species (*D'Ambrosio et al., 2008*; *Kranz et al., 2013*; *Kim et al., 2013*; *Yuen et al., 2017*). Condensin binding data reported to date has been collected from mitotic or interphase cells. Therefore, condensin binding patterns in terminally differentiated cells are unknown, and the binding patterns we observe may be unique to this cell type. Similarly, we saw that Cap-G binding was depleted at accessible chromatin and known enhancer and

regulatory element regions (*Leader et al., 2018*). This is contrary to a study in which Cap-G was shown to localise to active enhancers in human cancerous cells, promoting estrogen-dependent gene expression (*Li et al., 2015b*). However, conflicting evidence for condensin I binding at enhancer regions has been presented. For example, Barren has been shown to overlap with few known enhancers in mitotic *Drosophila* Kc167 cells (*Van Bortle et al., 2014*).

We observed that the majority of Cap-G binding in NSCs was depleted at accessible loci. This indicates that condensin may be involved in initiating or maintaining repressive chromatin states, and is supported by the observation that we see Cap-G most strongly associated with repressive chromatin states in NSCs. However, we also saw Cap-G association with some open chromatin regions, indicating that its role in regulation of gene expression may not be so clear. This is consistent the fact that we see de-repression of silenced genes in our RNA-seq data, as well as down-regulation of genes that are usually expressed. These data indicate that Cap-G does not have an exclusive repressive role. However, given that a large proportion of mis-expressed genes are not directly bound by Cap-G, we conclude that many of these changes are caused by indirect effects. Loss of Cap-D3 (condensin II) in cultured mitotic *Drosophila* cells results in spreading of repressive histone marks, affecting transcription of neighbouring genes, indicating that this complex may be required for maintaining boundaries between different chromatin environments (*Schuster et al., 2013*). It is conceivable that Cap-G has a similar function in post-mitotic neurons, which could help to explain the repression and de-repression of gene expression observed in our RNA-seq data.

The canonical role of condensin is as a molecular motor that extrudes DNA loops thereby organising chromosome architecture during mitotic chromosome compaction (*Paul et al., 2019*). Given that condensin has this ability to rearrange chromosome topologies, it is feasible that the action of Cap-G in neurons could be to remodel 3D chromatin structure for optimal gene expression. Since we see more severe consequences of Cap-G depletion in younger neurons, this may suggest that condensin is required to establish a mature post-mitotic chromosome conformation in fully differentiated neurons before synaptogenesis. Another SMC complex, cohesin, is well known to be required for the formation of topologically associated domains (*Yuen and Gerton, 2018*). Perturbation of cohesin in *Drosophila* neurons resulted in behavioural phenotypes and defects in axon pruning, also indicating that remodelling of chromatin architecture is likely to be important in neuronal maturation (*Pauli et al., 2008*).

## Differential requirement for Cap-G in newly born and mature neurons

We depleted Cap-G levels in two overlapping populations of cells, fully differentiated neurons (expressing *nSyb*), as well as immature to fully differentiated neurons (expressing *elav*). *elav-GAL4*-driven knockdowns consistently displayed more severe phenotypes than with *nSyb-GAL4* and had equally drastic changes in gene expression. This difference in phenotype can be attributed to the cells being targeted by the different drivers. *elav-GAL4* encompasses all neuronal cells, including newly born neurons that are in a more transient and plastic state, when compared to more mature, synapse forming-neurons targeted with *nSyb-GAL4*. This hints that Cap-G may serve a role in terminal differentiation of newly born neurons, but also maintenance of neuronal cell-state once matured. Consistent with this we saw much higher levels of Cap-G in newly born neurons than in more fully differentiated neurons, suggesting that Cap-G may play a more prominent role in the early life of the neuron.

Previous reports have stated that *elav-GAL4* may drive expression prematurely in embryonic NSCs (*Berger et al., 2007*). We made extensive efforts to determine whether *elav-GAL4* caused a reduction in Cap-G levels in NSCs or any defects in NSC division. Since we could not see any changes in Cap-G levels in NSCs and did not observe any phenotypes that we would expect to see from mitotic Cap-G knockdown, we concluded that the phenotypes we observed were a result of post-mitotic Cap-G depletion in neurons. However, the possibility remains that Cap-G knockdown may occur in an undetected subset of progenitor cells, or at levels too low to detect, that give rise to chromosome segregation related phenotypes. That phenotypes were observed with two independent neuronal GAL4 lines (*elav-GAL4* and *nSyb-GAL4*) provides further confidence in a *bona fide* role for Cap-G in neurons. In future, the development of GAL4 lines with even more precise spatial and temporal control will allow us to carefully interrogate the differences between condensin function in the different stages of neuronal maturation.

We initially decided to investigate Cap-G function in neurons based on a potential interaction with the neuronal transcription factor Lola-N. Lola-N is required in the early stages of neuronal differentiation to maintain the differentiated cell state (*Southall et al., 2014*). Similarly, we observed increased levels of Cap-G and more severe consequences from knockdown of Cap-G in the early stages of neuronal maturation. Therefore, it is intriguing to speculate that Cap-G/condensin I act together with Lola-N to maintain the neuronal transcriptome in differentiating neurons. Further studies will be necessary to verify whether there is in fact a functional relationship between these two proteins.

## Conclusion

In conclusion, we have shown that neuronal Cap-G is required for normal development and survival in *Drosophila*. This implicates the condensin I complex in a previously uncharacterised role in gene expression in post-mitotic cells. Further studies will be necessary to fully define the mechanism by which this regulation is mediated, whether that is by direct regulation of gene expression, or indirectly, through remodelling the topology of the 3D genome.

# Materials and methods

**Key resources table**

| Reagent type (species) or resource | Designation | Source or reference | Identifiers | Additional information |
|---|---|---|---|---|
| Gene (*D. melanogaster*) | *Cap-G* | | FLYB: FBgn0259876 | |
| Strain, strain background (*E. coli*) | DH5α | NEB | # C2987I | Competent cells |
| Genetic reagent (*D. melanogaster*) | P{GawB}elav$^{C155}$-GAL4 'elav-GAL4' | Bloomington *Drosophila* Stock Center | BDSC: #458 | |
| Genetic reagent (*D. melanogaster*) | w$^{1118}$; P{y$^{+t7.7}$ w$^{+mC}$ = GMR57 C10-GAL4}attP2 'nSyb-GAL4' | Bloomington *Drosophila* stock Center | BDSC: #39171 | |
| Genetic reagent (*D. melanogaster*) | w*;; P[w$^{+mC}$ = UAS-Nslmb-vhhGFP4] 'UAS-deGradFP' | Bloomington *Drosophila* stock Center | BDSC: #38421 | *Caussinus et al., 2012* |
| Genetic reagent (*D. melanogaster*) | Cap-G$^{EGFP}$ | *Kleinschnitz, 2020* | | Stefan Heidmann Lab |
| Genetic reagent (*D. melanogaster*) | barren$^{EGFP}$ | *Kleinschnitz, 2020* | | Stefan Heidmann Lab |
| Genetic reagent (*D. melanogaster*) | wor-GAL4; tub-GAL80$^{ts}$ | Southall Lab | | Southall Lab stocks |
| Genetic reagent (*D. melanogaster*) | tub-GAL4 | Bloomington *Drosophila* stock Center | | |
| Genetic reagent (*D. melanogaster*) | UAS-LT3-Cap-G-Dam | This paper | | See Materials and methods |
| Genetic reagent (*D. melanogaster*) | UAS-LT3-Dam | Southall Lab, *Southall et al., 2013* | | |
| Antibody | Anti-GFP (chicken-polyclonal) | Abcam | ab13970 | IF(1:2000) |
| Antibody | Anti-Lola-N (rabbit- | Southall Lab, *Southall et al., 2014* | | IF(1:10) |
| Antibody | Anti-elav (c) (rat- monoclonal) | DSHB | | IF(1:500) |
| Antibody | Anti-deadpan (guinea pig) | Brand Lab | | Donated by Andrea Brand Lab IF(1: 10000) |
| Antibody | Anti-pH3 (rabbit-polyclonal) | Millipore | #06–570 | IF(1:500) |

*Continued on next page*

*Continued*

| Reagent type (species) or resource | Designation | Source or reference | Identifiers | Additional information |
|---|---|---|---|---|
| Antibody | Anti-γ-H2AV (mouse-monoclonal) | DSHB | Cat name: UNC93-5.2.1 | IF(1:200) |
| Recombinant DNA reagent | Plasmid-*pUAST-attB* | *Bischof et al., 2007* | | |
| Sequence-based reagent | Cap-G_FW | This paper | | TGGTACCGCATAAT AACATGGCCAAACCAAAG |
| Sequence-based reagent | Cap-G_RV | This paper | | GCGATTTTTCTTCAT CAGATCCTCTTCAGAG ATGAGTTTCTGTTCTTT CCTCCTGCTGCG |
| Sequence-based reagent | Dam_FW | This paper | | GCGCAGCAGGAGG AAAGAACAGAAACTCA TCTCTGAAGAGGATCTG ATGAAGAAAAATCGC |
| Sequence-based reagent | Dam_RV | This paper | | TTCACAAAGATCCTC TAGAGGTACCCTCGATT AACCGGCTTTTTTCGC GGGTGAAACGACTCC |
| Commercial assay or kit | Click-iT TUNEL Alexa Fluor 549 | Thermo Fisher | Cat no: C10618 | |
| Software, algorithm | R, R Studio | R 3.4.3 | | |
| Software, algorithm | Prism 8 | Graphpad Prism eight for Windows | | |
| Other | DAPI (DNA stain) | Thermo Fisher | #D1306 | Concentration: 1:20000 |

## Fly stocks

Cap-G[EGFP] and barren[EGFP] CRISPR knock-in lines were generated by first inserting cassettes directing expression of EGFP in the eye, immediately downstream of the *Cap-G* and *barren* reading frames within the context of their genomic loci. This facilitated easy screening of knock-in individuals due to their green eye fluorescence. Upon FLP-recombinase mediated excision of the eye-specific promotor, eye fluorescence was lost and at the same time a continuous reading frame was generated between *Cap-G* or *barren* and *EGFP*. Thus, C-terminally EGFP-fused Cap-G and Barren variants are expressed under control of their genomic regulatory elements. Further details of the strain construction are described elsewhere (*Kleinschnitz, 2020*). For degradFP experiments we used the line *w\*;; P[w[+mC] = UAS-Nslmb-vhhGFP4]* (*Caussinus et al., 2012*). *tubulin-GAL4/TM6B* was used for ubiquitous degradation of Cap-G. *UAS-mcd8-GFP* was used for fluorescent reporter experiments. Neuron-specific Cap-G knockdown was driven by the following GAL4 lines *P(GawB)elav[C155] -GAL4* for newly born neurons and *w[1118]; P{y[+t7.7] w[+mC] = GMR57 C10-GAL4}attP2* (*nSyb-GAL4* - Bloomington #39171) for mature neurons. *wor-GAL4* was used to drive expression in NSCs (*Albertson et al., 2004*).

## Immunohistochemistry and in-situ hybridisation

Third instar larvae and adult CNS were dissected in 1x PBS and tissue was fixed for 20 min in 4% formaldehyde (Polysciences, Inc, 10% methanol-free) diluted in PBST (0.3% Triton X-100 in PBS). Tissue washes were done in PBST every 5–15 min. Normal Goat Serum (2% in PBST) was used for tissue blocking (RT, 15 min- 1 hr) and subsequent overnight primary antibody incubation. Tissue was mounted on standard glass slides in Vectashield Mounting medium (Vector laboratory).

Embryos were kept at 25°C and collected every 12–15 hr. Embryos were prepared for confocal imaging using a standard protocol as previously described (*Southall et al., 2014*). Embryos were dechorionated using 50% bleach solution and subsequently fixed in a 1:1 solution of 4% formaldehyde to Heptane. After fixation embryos were washed in 100% methanol to ensure removal of the

vitelline membrane and subsequently washed with PBST. Tissue was blocked in 2% NGS (RT, 15 min- 1 hr) and then incubated with appropriate primary antibodies overnight.

Primary antibodies used include: rat anti-elav 1:500 (Developmental Studies Hybridoma Bank, DSHB), chicken anti-GFP 1:2000 (Abcam), rabbit anti-phospho-Histone H3 Ser10 (pH3) 1:500 (Merck Millipore, 06–570), guinea pig anti-deadpan 1: 1000 (provided by A. Brand), rabbit anti-Lola-N 1:10 (*Southall et al., 2014*) and mouse anti-ɣ-H2Av 1:500 (DSHB - *Lake et al., 2013*). Secondary antibodies used include Alexa Fluor 488, 545 and 633 1:200 (Life technologies) and tissue was incubated for 1.5 hr at room temperature. The DNA stain DAPI (1: 20,000) was used as nuclear counterstain.

TUNEL staining on embryos was performed using the Click-iT TUNEL Alexa Fluor 549 (Thermo-Fisher). The provided protocol was followed. The only optimisation included incubation of fixed embryos with Proteinase K for 30 min at room temperature and subsequent post-fixation in 4% formaldehyde for 15 min at room temperature.

For in-situ hybridisation experiments, RNA probes were designed using LGC Biosearch Technologies' Stellaris RNA FISH Probe Designer against *Cap-G* exon four and with Quasar 570 dye. Tissue was treated following the published protocol (*Yang et al., 2017*). Fixed and blocked tissue was incubated in hybridisation buffer with *Cap-G* probe (3 µM) and primary antibody of interest for 8–15 hr.

## Imaging and image analysis

Samples were imaged using a confocal microscope Zeiss LSM 510 and Leica CF8. Analysis of acquired images was done using the Fiji (*Schindelin et al., 2012*) and Icy-Bioimage Analysis software (*de Chaumont et al., 2012*). Icy plugin Spot Detector (*Olivo-Marin, 2002*) was used to analyse the total number of cells when quantifying dividing NSCs, pH3 staining, DNA damage (ɣ-H2Av) as well as TUNEL staining, filters were adjusted per experiment and kept constant across control and experimental images. For nuclear area analysis the Ilastik software was used for bulk image segmentation, (*Berg et al., 2019*) to recognise each nucleus as an individual ROI. Subsequently, the segmented image was analysed in Fiji using Analyze Particles and extracting the mean nuclei area (>1000 cells / replicate) in a total of 7 embryonic replicates. To quantify live GFP levels we took a z-stack (four slices minimum of the VNC) per embryo. Actively dividing cells displayed highest GFP levels, therefore binary images were created by using a threshold (Outsu's thresholding) to select dividing cells as regions of interest (ROI). Mean pixel intensity per ROI (dividing cell) was calculated using Analyze Particles plugin in Fiji (>30 cells/ embryo). Average pixel intensity of ROI was calculated across z-stack slices, for a total of 10 biological replicates. To quantify the GFP levels in NSCs, images were analysed from three biological replicates per condition. The Dpn positive cells were selected as individual ROIs as described above. The average pixel intensity detected for GFP and for Dpn were extracted using the Analyze Particles plugin in Fiji (>70 cells / embryo). The GFP/Dpn ratio of pixel intensity was calculated per cell. Statistical significance was analysed using a Nested t-test.

## Behavioural and phenotypic assays

All animals were kept at 25°C unless otherwise stated. For the developmental survival assay three biological replicates per genotype were used. Flies were allowed to lay for 5 hr and then 100 embryos per replicate were collected. After 24 hr, L1 larvae were transferred to a food vial and allowed to develop. The number of pupae (5 days post- lay) and eclosed adults (10 days post-laying) were recorded. Adult survival assays were performed on three replicates, with 20 flies per replicate. Male and female animals were separated upon eclosion. Death was scored daily until total number of flies deceased.

Behavioural assays were performed as previously described (*Nichols et al., 2012*). Locomotion assay was performed on 3rd instar larvae using 10 biological replicates and three technical replicates. Individual larvae were setup in clear agar plates against a 0.5 cm$^2$ grid. Larvae were left to acclimatise for 1 min and locomotion was recorded as distance covered per minute. Negative geotaxis assays were performed on 10 male flies and 10 technical replicates per genotype. Flies were set up in climbing vials against a grid and a camera. Video recording was started before sharply tapping flies to the bottom of vials. Videos were analysed using Icy, the frame in which the first fly reached the top of the vial was extracted. The coordinates of each fly were used to calculate the average distance climbed per replicate. Statistical tests and plots were performed using the software GraphPad Prism version eight for Windows.

## Targeted DamID

We generated a *UAS-Cap-G-Dam* line to use in Targeted DamID experiments. The isoform Cap-G-PF was amplified by PCR from cDNA library. The *Dam* construct was fused on the C-terminus of Cap-G using fusion PCR. The *Dam* sequence is frequently fused to the N-terminus of the protein of interest, however, the fusion of a protein to the Cap-G C-terminus has been previously shown to have no effect on protein integrity or function (*Herzog et al., 2013*). The *mCherry* sequence used for the primary ORF in the TaDa cassette was amplified from pUAST-LT3-Dam (*Southall et al., 2013*) template and fused by PCR to the N-terminal of *Cap-G-Dam* construct. Finally the *mCherry-Cap-G-Dam* construct is cloned into the pUAST-attB (*Bischof et al., 2007*) plasmid using Gibson Assembly (*Gibson et al., 2010*).

To infer cell-specificity in our Targeted DamID (*Southall et al., 2013*) experiments we used the following driver lines: *wor-GAL4; tub-Gal80ts* for neuronal stem cells, *elav-GAL4; tub-GAL80ts* for immature and mature neurons and *tub-GAL80ts; nSyb-GAL4* for mature neurons. The *UAS-Cap-G-Dam* line was used to profile Cap-G binding and the Dam-only line *tub-GAL80ts; UAS-LT3-NDam* (*Southall et al., 2013*) was used as a control for Dam expression and Chromatin Accessibility TaDa experiments (*Aughey et al., 2018*).

Animals were crossed to the desired driver line and embryos collected at 25°C for 4 hr. To obtain third instar larval brains, embryos were raised at 18°C for seven days post collection and subsequently placed at 29°C for 24 hr to induce Dam- expression. 30 larval brain per replicate were dissected in PBS with 100 mM EDTA. Extraction of Dam-methylated DNA and genomic libraries were performed as previously described (*Marshall et al., 2016*). Illumina HiSeq single-end 50 bp sequencing was performed on two biological replicates. Sequencing obtained was mapped to release 6.03 of the *Drosophila melanogaster* genome and normalised against *Dam-only* control data (*Marshall et al., 2016*).

## Peak calling and annotation

Significant peaks were called and mapped to genes using a custom Perl program that allows for the identification of broadly bound regions (available at https://github.com/tonysouthall/Peak_calling_DamID; copy archived at https://github.com/elifesciences-publications/Peak_calling_DamID; *Southall, 2019*). In brief, a false discovery rate (FDR) was calculated for peaks (formed of two or more consecutive GATC fragments) for the individual replicates. Then each potential peak in the data was assigned an FDR. Any peaks with less than a 0.01% FDR were classified as significant. Significant peaks present in both replicates were used to form a final peak file. Any gene (genome release 6.11) within 5 kb of a peak (with no intervening genes) was identified as a potentially regulated gene. (*Marshall et al., 2016*) Cap-G peaks were assigned to genomic features using HOMER software annotatePeaks pipeline (*Heinz et al., 2010*).

## CATaDa and chromatin accessibility analysis

CRM data were extracted from the REDfly database (*Rivera, 2018*; http://redfly.ccr.buffalo.edu/index.php). Coordinates were converted to BED format for comparison to TaDa data. For CATaDa data analysis Dam only reads were mapped to release 6.22 of the *Drosophila* genome and binned to GATC regions using a previously described pipeline (*Aughey et al., 2018*) (available at: https://github.com/tonysouthall/damidseq_pipeline_output_Dam-only_data; *Marshall and Brand, 2015*). Processed CATaDa data were normalised to total number of reads per million before comparison to other data types. Regions of open chromatin defined in *Figure 4E* were defined by calling peaks on CATaDa data using a previously described pipeline (available at https://github.com/tonysouthall/Peak_Calling_for_CATaDa; *Southall, 2017*).

## RNA-seq and data analysis

RNA-seq was performed on three biological replicates, for Cap-G knockdown and controls respectively. Control genotypes were *elav-GAL4; UAS-deGradFP* and *nSyb-GAL4; UAS-deGradFP* for *elav-KD* and *nSyb-KD* respectively. Total RNA was extracted from dissected 3$^{rd}$ instar larvae CNS, 35 per replicate, using a standard TRIzol extraction protocol (*Thermo Fisher Scientific, 2016*). RNA library preparation and sequencing was performed by Beijing Genomics Institute (BGI).

Sequencing data was mapped to the *Drosophila* genome (release 6.22) using STAR (*Dobin et al., 2013*). Mapped files were collected in a matrix using *featureCounts* from the Rsubread package (*Liao et al., 2019*). Differential expression analysis and MA plots were carried out using the Deseq2 R package (*Love et al., 2014*). Volcano plots were generated using EnhancedVolcano R package (*Blighe, 2019*). Genes that had an adjusted p-value<0.05 and a $\log_2$ fold change greater than 1 (for upregulated) or less than $-1$ (for downregulated) were classified as significant. Heatmap generated using pheatmap package in R.

Tissue expression data from FlyAtlas 2 was used to determine tissue of origin of upregulated genes (*Leader et al., 2018*). Larval FPKM values above two were considered non-background and were used as a threshold for tissue specificity. Expression data for a list of genes of interest was extracted from the FlyAtlas 2 database and data analysis carried out using Pandas in Python (*McKinney, 2010*) using a custom code (available at: https://github.com/amh111/FlyAtlas2Scraper) copy archived at https://github.com/elifesciences-publications/FlyAtlas2Scraper; *Hassan, 2020*).

## Statistical analysis and Gene Ontology

Overlap analysis of peak-files from different datasets was done using Bedtools Intersect and statistical significance determined by Fisher's exact test using Bedtools fisher (*Quinlan and Hall, 2010*). Deeptools was used to generate average signal profiles, heatmaps, principal component analysis and correlation matrices (*Ramírez et al., 2016*). Enrichment GO analysis was performed on gene lists of interest using the R package clusterProfiler (*Yu et al., 2012*). All other figures were produced using the ggplot2 package in R.

# Acknowledgements

We would like to thank the Southall lab and Imperial College fly community for help and feedback on this project. We thank Mareike Jordan, Kristina Kleinschnitz and Nina Viessmann for generating and characterizing the Cap-G-EGFP and Barren-EGFP CRISPR knock-in lines. We are particularly grateful to Alicia Estacio-Gomez, Owen Marshall, and Seth Cheetham for critical reading of this manuscript. We would like to thank Andrea Brand for generously providing antibodies used in these experiments. For fly stocks, we thank the Bloomington *Drosophila* Stock Center (NIH P40OD018537). The Facility for Imaging by Light Microscopy (FILM) at Imperial College London is part-supported by funding from the Wellcome Trust (grant 104931/Z/14/Z) and BBSRC (grant BB/L015129/1). This work was funded by a Wellcome Trust Investigator grant 104567 to TDS, a BBSRC grant BB/P017924/1 to TDS and GNA, and a BBSRC 1+3 DTP studentship BB/M011178/1 (Amira Hassan).

# Additional information

### Funding

| Funder | Grant reference number | Author |
|---|---|---|
| Wellcome | 104567/Z/14/Z | Tony D Southall |
| Biotechnology and Biological Sciences Research Council | BB/P017924/1 | Gabriel N Aughey Tony D Southall |
| Biotechnology and Biological Sciences Research Council | BB/M011178/1 | Amira Hassan |
| Deutsche Forschungsgemeinschaft | HE2354/23-2 | Stefan K Heidmann |
| Deutsche Forschungsgemeinschaft | HE2354/4-1 | Stefan K Heidmann |

The funders had no role in study design, data collection and interpretation, or the decision to submit the work for publication.

## Author contributions
Amira Hassan, Data curation, Formal analysis, Investigation, Visualization, Methodology, Writing - original draft, Writing - review and editing; Pablo Araguas Rodriguez, Data curation, Investigation, Methodology, Writing - review and editing; Stefan K Heidmann, Resources, Writing - review and editing; Emma L Walmsley, Investigation, Writing - review and editing; Gabriel N Aughey, Conceptualization, Data curation, Formal analysis, Supervision, Investigation, Visualization, Methodology, Writing - review and editing; Tony D Southall, Conceptualization, Supervision, Funding acquisition, Visualization, Methodology, Project administration, Writing - review and editing

## Author ORCIDs
Amira Hassan https://orcid.org/0000-0003-1640-1602
Emma L Walmsley http://orcid.org/0000-0002-3785-075X
Gabriel N Aughey https://orcid.org/0000-0001-5610-9345
Tony D Southall https://orcid.org/0000-0002-8645-4198

## Decision letter and Author response
Decision letter https://doi.org/10.7554/eLife.55159.sa1
Author response https://doi.org/10.7554/eLife.55159.sa2

## Additional files
### Supplementary files
• Transparent reporting form

### Data availability
All raw sequence files and processed files have been deposited in the National Center for Biotechnology Information Gene Expression Omnibus (accession number GSE142112).

The following dataset was generated:

| Author(s) | Year | Dataset title | Dataset URL | Database and Identifier |
|---|---|---|---|---|
| Southall TD, Hassan A, Aughey GN | 2019 | Characterisation of Cap-G binding and its role in post-mitotic neurons in Drosophila | https://www.ncbi.nlm.nih.gov/geo/query/acc.cgi?acc=GSE142112 | NCBI Gene Expression Omnibus, GSE142112 |

The following previously published datasets were used:

| Author(s) | Year | Dataset title | Dataset URL | Database and Identifier |
|---|---|---|---|---|
| Marshall OJ, Brand AH | 2017 | Chromatin state changes during neural development revealed by in vivo cell-type specific profiling | https://www.ncbi.nlm.nih.gov/geo/query/acc.cgi?acc=GSE77860 | NCBI Gene Expression Omnibus, GSE77860 |
| Leader DP, Krause SA, Pandit A, Davies SA, Dow JA | 2018 | FlyAtlas 2: a new version of the Drosophila melanogaster expression atlas with RNA-Seq, miRNA-Seq and sex-specific data. | https://www.ebi.ac.uk/ena/data/view/PRJEB22205 | European Nucleotide Archive, PRJEB22205 |

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
