## [Decision Letter]

**Acceptance summary:**

In this manuscript the authors identify functions in postmitotic neurons for the chromatin structural protein CapG, which is part of the condensin I complex. Condensin is well known to play a role in chromosome condensation and loop extrusion in recently divided cells. However its functions in postmitotic cells have been less clear. The idea that nuclear architectural proteins might be repurposed in non-dividing cells for transcriptional regulatory purposes is interesting, and this manuscript provides a solid set of data for establishing that there are non-mitotic function of CapG in neurons.

**Decision letter after peer review:**

Thank you for submitting your article "Condensin I subunit Cap-G is essential for proper gene expression during the maturation of post-mitotic neurons" for consideration by *eLife*. Your article has been reviewed by three peer reviewers, one of whom is a member of our Board of Reviewing Editors, and the evaluation has been overseen by K VijayRaghavan as the Senior Editor. The reviewers have opted to remain anonymous.

The reviewers have discussed the reviews with one another and the Reviewing Editor has drafted this decision to help you prepare a revised submission.

Summary:

In this manuscript the authors identify functions in postmitotic neurons for the chromatin structural protein CapG, which is part of the condensin I complex. Condensin is well known to play a role in chromosome condensation and loop extrusion in recently divided cells. However its functions in postmitotic cells have been less clear. The authors became interested in CapG in neurons when they found it binding a neuronal transcription factor by yeast two-hybrid. Here they show CapG is expressed in postmitotic neurons and then studied phenotypes in flies where CapG was knocked out in neurons and used the TaDa DamID method to study CapG binding in neurons.

All three reviewers found the idea that nuclear architectural proteins might be repurposed in non-dividing cells for transcriptional regulatory purposes to be compelling, and they felt that this manuscript provides a solid set of data for establishing that there are non-mitotic function of CapG in neurons. However there were several significant technical concerns raised, some of which may be addressed with text revisions, but in particular the interpretation of TaDa data require additional validation.

Essential revisions:

Essential revisions fall into three areas relating to 1) the TaDa peak calling, 2) the relationship of CapG binding to chromatin and gene expression, 3) the interpretation of the elav and *nsyb* phenotypes.

1) Although the reviewers agreed that the TaDa approach was a good one for mapping CapG binding in neurons, all expressed some concerns about the details of the TaDa data. The reviewers noted that peak calling using the TaDa approach is challenging since the signal to noise is very modest compared to high quality ChIP-seq signals. Furthermore for CapG the signals appear to be very broad, which makes them poorly amenable to traditional peak calling. The reviewers agreed that the authors should either systematically validate their peak calling of TaDa datasets with replicates as they have done in a separate *ELife* paper published recently, or adapt their computational analyses to reflect the broader binding patterns of CapG compared to transcription factors. This was a topic of substantial discussion among the reviewers during the commentary period. Specific comments from the reviewers on this point are below. We include this information so the authors can see the kinds of concerns that were raised and we encourage them to offer what they feel is the strongest response either with the addition of more data or with a revised analysis approach.

• Figure 4A: What fraction of TaDa peaks are statistically detected in all replicates for each condition? Reproducibility between replicates is important for understanding how much of the differences between conditions is due to biological or technical variability. Could the author please show the individual replicates for all examples in 4A? Rigorous analyses using multiple replicates (as the authors performed in Sen et al. e*Life* 2019) would be necessary.

• Alternatively, using "aggregate" computational analyses such as in Figure 4D-F would be acceptable, if the peak analyses proved to be not reproducible across replicates for individual genomic loci. I would leave the biological replicate experiments up to the authors, who may find it useful for statistical analyses comparing the genetic lines and possibly to strengthen/convince us that their peak calling is the accurate way to characterize CapG binding.

• Condensin may not bind in the same manner to chromatin as transcription factors or histone modifications, which often show sharp peaks. Based on the examples presented in Figure 4A, the observed negative-going "valleys" may be also meaningful signal. Using the aggregate Cap-G TaDa signal across the entire gene body might be yet another alternative statistical method for comparing between conditions and across replicates (this is a common strategy for RNA-Seq statistical analyses).

• Looking at their few examples, it appears CapG binds more like chromatin regulators that have broad gene-body distributions (i.e. elongation regulators, MECP2). Peak callers like MACS2 used in this study would not be appropriate in this scenario and alternative computational methods are more appropriate. If the authors wish to use peak calling, they must use replicates with statistical measures to validate peak calling reproducibility.

• In the examples in 4A, it is unclear how to interpret the results. For example, for the dacapo gene, the peaks upstream of the gene appear preserved in all experiments (even if peak calling may be different between conditions due to statistical thresholds). However, there is no mention of these upstream peaks. Another example are the peaks identified in the genebody of unc-89, which are somewhat randomly distributed across the various conditions. Are the differences between conditions biological or due to technical variability? Finally, for the bruchpilot gene, the authors state that Cap-G binding is stronger in NSCs than mature neurons. However, there are several positive signals along the gene body of bruchpilot that are conserved in both the elav-GAL4 and *nsyb*-GAL4 neurons and, although distinct, appear to have equal amplitude as the "statistically-significant" peaks in the wor-GAL4 condition. Together, if these examples represent the typical peak calling, this reviewer is concerned about interpretability of subsequent analyses that rely on peak calling (4B, 4C).

Finally, one reviewer also raised an independent point about these data, saying: "The DamID data suggest there are different binding patterns of CapG in NSCs, immature neurons and mature neurons. Might there be differences in expression in the different lines that could make apparent differences in binding? Also are the DamID lines functional overexpressors of CapG? If so, how do the authors know the binding of the transgene is representative of the endogenous gene?"

2) A second major concern raised by the reviewers regards the comparison of CapG binding to open chromatin and gene expression.

For Figure 4D-E, the methods for how the authors identified putative enhancers or open chromatin is missing (authors only indicate from REDFly). Are these putative enhancers and open chromatin curated from all cells or matched to the specific cell types and developmental stages they are studying? If the vast majority of enhancers and "open" chromatin regions are largely inactive in bulk populations of cells, then the observed results would reflect Cap-G preferentially binding to open chromatin, but only in cell type-specific manners. The authors should perform the same analyses using active enhancers and bonafide open chromatin regions specifically from their tested cell stages. If they have done so, they should clearly state how they did this.

With respect to how this relates to gene expression, the reviewers were not convinced by the arguments in the text. How do the authors propose that CapG bind only to repressive chromatin, but upon knockdown of CapG, then downregulated genes are localized to active chromatin? The authors need to comment on this, particularly since they are arguing CapG is directly regulating both the upregulated and downregulated genes via its binding to both groups of genes. As another reviewer stated, It is unclear from their data why equal numbers of genes are up and down regulated. If Cap-G preferentially binds to repressive chromatin, as the authors argue in Figure 4, how does loss of Cap-G leads to the downregulation of neuronal genes, which should be regulated by active chromatin in neurons? I think that the authors should comment on: a) the low number of overlapping genes up- and down-regulated in *nsyb*- and elav-driven CapG knockouts and b) on the huge difference between the up- and down-regulated genes between the 2, which could be very interesting.

3) The reviewers raised concerns that elav can be expressed in progenitors and thus felt that the authors cannot use the elav line as definitive evidence for postmitotic functions of CapG.

They indicated that addition of a dividing cell marker (such as Dpn) in Figure 1—figure supplement 1A would be needed to validate that CapG is not reduced in dividing cells. Given that the authors show convincing behavioral data using the *nsyb*-KD line that indicates a function for Condensin in neurons, the conclusions based on the elav line could be toned down to admit possible expression in progenitors without undermining the story that the authors with to tell here.

With respect to the behavioral phenotypes in the *nsyb* line, although the reviewers were convinced that there were phenotypes (confirming functions of CapG in neurons) they did not think that detailed description of these phenotypes was highly meaningful given the survival deficits. Thus the commentary on CapG functions in behavior could also be written back.

---

## [Author Response]

Essential revisions:Essential revisions fall into three areas relating to 1) the TaDa peak calling, 2) the relationship of CapG binding to chromatin and gene expression, 3) the interpretation of the elav and nsyb phenotypes.1) Although the reviewers agreed that the TaDa approach was a good one for mapping CapG binding in neurons, all expressed some concerns about the details of the TaDa data. The reviewers noted that peak calling using the TaDa approach is challenging since the signal to noise is very modest compared to high quality ChIP-seq signals. Furthermore for CapG the signals appear to be very broad, which makes them poorly amenable to traditional peak calling. The reviewers agreed that the authors should either systematically validate their peak calling of TaDa datasets with replicates as they have done in a separate ELife paper published recently, or adapt their computational analyses to reflect the broader binding patterns of CapG compared to transcription factors. This was a topic of substantial discussion among the reviewers during the commentary period. Specific comments from the reviewers on this point are below. We include this information so the authors can see the kinds of concerns that were raised and we encourage them to offer what they feel is the strongest response either with the addition of more data or with a revised analysis approach.• Figure 4A: What fraction of TaDa peaks are statistically detected in all replicates for each condition? Reproducibility between replicates is important for understanding how much of the differences between conditions is due to biological or technical variability. Could the author please show the individual replicates for all examples in 4A? Rigorous analyses using multiple replicates (as the authors performed in Sen et al. eLife 2019) would be necessary.

We thank the reviewers for bringing this matter to our attention. In our original submission we neglected to provide a full explanation of our peak calling process. Furthermore the annotations of peaks on Figure 4A were incorrect due to an error in processing these figures. Therefore, some aspects of these figures were misleading.

The peak calling programme that we have used is specifically designed to be able to detect the broad peaks produced by DamID data, in which the resolution is limited by the occurrence of GATC fragments. Using this programme a peak is only called if it was deemed to be significant in all biological replicates. Therefore the peaks called should indicate regions that have good agreement between replicates.

Overall, we find that reproducibility between replicates is very good. To illustrate this we have provided some further analysis (Figure 4—figure supplement 1) highlighting the good correlations between replicates. We have also updated Figure 4A to highlight the reproducibility. These updated figures now also include the correct peak annotations.

• Alternatively, using "aggregate" computational analyses such as in Figure 4D-F would be acceptable, if the peak analyses proved to be not reproducible across replicates for individual genomic loci. I would leave the biological replicate experiments up to the authors, who may find it useful for statistical analyses comparing the genetic lines and possibly to strengthen/convince us that their peak calling is the accurate way to characterize CapG binding.

For the TaDa experiments, we have performed two biological replicates, for which there is good reproducibility (please see response above). We have also added a new figure showing the aggregate binding of Cap-G (Figure 4—figure supplement 1C).

• Condensin may not bind in the same manner to chromatin as transcription factors or histone modifications, which often show sharp peaks. Based on the examples presented in Figure 4A, the observed negative-going "valleys" may be also meaningful signal. Using the aggregate Cap-G TaDa signal across the entire gene body might be yet another alternative statistical method for comparing between conditions and across replicates (this is a common strategy for RNA-Seq statistical analyses).

It is not unusual to observe “negative” peaks in DamID data. These occur as a result of the normalisation against the Dam only control. Since the Y-axes display the ratio of Cap-G-Dam / Dam-only, negative values are expected. At loci which display high levels of background methylation but no association with the dam-fusion, a negative peak will be produced. Therefore, the negative peaks may not meaningfully be interpreted as representing bona-fide troughs in Cap-G binding relative to zero. However, they do give some indication of the chromatin accessibility at these loci, which is the basis for our CATaDa technique which we talk about in more detail in response to another of the reviewers’ points. We have added a note in the legend of Figure 4 highlighting this detail.

We have conducted the analysis that the reviewers suggest regarding the aggregated signal across gene bodies. In agreement with our peak annotation presented in figure 4C, we find that Cap-G signal is most strongly enriched across gene bodies whilst being depleted from TSS and TES. These new data are provided in Figure 4—figure supplement 1C.

• Looking at their few examples, it appears CapG binds more like chromatin regulators that have broad gene-body distributions (i.e. elongation regulators, MECP2). Peak callers like MACS2 used in this study would not be appropriate in this scenario and alternative computational methods are more appropriate. If the authors wish to use peak calling, they must use replicates with statistical measures to validate peak calling reproducibility.

As previously stated, we believe this misconception has arisen from a failure on our part to adequately explain the computational methods employed to analyse Cap-G peaks. Since our pipeline is suitable for analysis of broad peaks by only considering multiple adjacent enriched peaks, we believe that it is suitable for the analysis of this dataset. As we mentioned in response to a previous comment, our peak calling only considers peaks present between replicates, and statistical measures are employed to ensure the fidelity of this analysis. The text has been updated to ensure that these details are more carefully highlighted.

• In the examples in 4A, it is unclear how to interpret the results. For example, for the dacapo gene, the peaks upstream of the gene appear preserved in all experiments (even if peak calling may be different between conditions due to statistical thresholds). However, there is no mention of these upstream peaks. Another example are the peaks identified in the genebody of unc-89, which are somewhat randomly distributed across the various conditions. Are the differences between conditions biological or due to technical variability? Finally, for the bruchpilot gene, the authors state that Cap-G binding is stronger in NSCs than mature neurons. However, there are several positive signals along the gene body of bruchpilot that are conserved in both the elav-GAL4 and nsyb-GAL4 neurons and, although distinct, appear to have equal amplitude as the "statistically-significant" peaks in the wor-GAL4 condition. Together, if these examples represent the typical peak calling, this reviewer is concerned about interpretability of subsequent analyses that rely on peak calling (4B, 4C).

In our original submission the presentation of these data and previously described problems with peak annotation on Figure 4A made these data difficult to interpret. We have updated this figure to include better illustrative examples of the points we are trying to convey (with correctly positioned peaks indicated). In the new version we have included genes that more unambiguously display changes in binding between conditions. We have also altered the way called peaks are highlighted on the figure which should make the figure easier to interpret. We have also altered the text to reflect these changes and expand on the interpretation of this figure. Furthermore, we have added a supplementary figure with further illustrative examples (Figure 4—figure supplement 1D).

Finally, one reviewer also raised an independent point about these data, saying: "The DamID data suggest there are different binding patterns of CapG in NSCs, immature neurons and mature neurons. Might there be differences in expression in the different lines that could make apparent differences in binding? Also are the DamID lines functional overexpressors of CapG? If so, how do the authors know the binding of the transgene is representative of the endogenous gene?"

There are several reasons to suspect that the relative expression levels of the drivers do not have a significant effect on the binding observed. Firstly, the Dam-only controls were expressed using the same drivers as the Cap-G Dam fusion. Since the Cap-G-Dam data are normalised against the dam-only data, this controls for differences in expression levels.

The primary open reading frame used to attenuate Dam translation is highly effective. It has previously been shown that the levels of Dam protein produced are so low as to be completely undetectable by western blot when expressed from a tubulin promoter (Southall et al., 2013). This is a requirement for DamID to avoid saturating methylation and toxic effects. Therefore, it is unlikely that expression of the Cap-G-Dam fusion will result in overexpression phenotypes since the levels produced are so low.

We are confident that the binding profiles produced are representative of the binding of the endogenous protein since Cap-G is tagged on the same terminus as in the Cap-G-EGFP line used in this study. This line is homozygous viable indicating that Cap-G is able to tolerate a fusion protein at this position and continue to fulfil its usual function. Since Dam is a small protein (comparable in size to GFP), it is very unlikely that it interferes with Cap-G localisation.

2) A second major concern raised by the reviewers regards the comparison of CapG binding to open chromatin and gene expression.For Figure 4D-E, the methods for how the authors identified putative enhancers or open chromatin is missing (authors only indicate from REDFly). Are these putative enhancers and open chromatin curated from all cells or matched to the specific cell types and developmental stages they are studying? If the vast majority of enhancers and "open" chromatin regions are largely inactive in bulk populations of cells, then the observed results would reflect Cap-G preferentially binding to open chromatin, but only in cell type-specific manners. The authors should perform the same analyses using active enhancers and bonafide open chromatin regions specifically from their tested cell stages. If they have done so, they should clearly state how they did this.

The data indicated in Figure 4E are experimentally validated cis-regulatory modules (CRMs) downloaded from the REDfly database. This dataset includes all CRMs without filtering for specific cell-types or stages. We chose to use this broad dataset initially to provide an overview of Cap-G binding at functional elements in the genome. These data indicate that Cap-G binding is depleted at these loci, however, the reviewer is correct in their claim that binding at open chromatin may be masked by lack of cell-specificity. Previously we showed that by re-analysing the Dam-only data against which the Dam-fusion protein is normalised, we are able to obtain data reflecting the chromatin accessibility of individual cell-types since the untethered dam is able to freely methylate all accessible regions (Aughey et al., 2018). We referred to this technique as CATaDa. In Figure 4E we have utilised these control Dam-only data from our TaDa experiments to compare binding to areas of accessible chromatin in the relevant cell-type – indicating that Cap-G binding is indeed depleted at cell-specific accessible loci. We realise now that we did not describe this approach in the text in much detail and neglected to outline the analysis of these data in our Materials and methods. In the revised version we have expanded on these details in the Results and Materials and methods section to make clear the data-types used and the analyses undertaken.

With respect to how this relates to gene expression, the reviewers were not convinced by the arguments in the text. How do the authors propose that CapG bind only to repressive chromatin, but upon knockdown of CapG, then downregulated genes are localized to active chromatin? The authors need to comment on this, particularly since they are arguing CapG is directly regulating both the upregulated and downregulated genes via its binding to both groups of genes. As another reviewer stated, It is unclear from their data why equal numbers of genes are up and down regulated. If Cap-G preferentially binds to repressive chromatin, as the authors argue in Figure 4, how does loss of Cap-G leads to the downregulation of neuronal genes, which should be regulated by active chromatin in neurons? I think that the authors should comment on: a) the low number of overlapping genes up- and down-regulated in nsyb- and elav-driven CapG knockouts and b) on the huge difference between the up- and down-regulated genes between the 2, which could be very interesting.

Whilst we see a strong enrichment of Cap-G at repressive chromatin types, and depletion at open chromatin, these data are not sufficient to conclude that the function of Cap-G results solely in gene repression. Furthermore, whilst binding of Cap-G is mostly depleted at sites of open chromatin, there appears to be a subset of accessible loci which Cap-G does associate with. This can be observed in the heatmaps of Figure 4E (dark blue at tops of heatmaps indicating strong signal in minority of regions). Therefore, since we see Cap-G associating mostly with repressive and inaccessible chromatin, we conclude that it may be involved mostly with gene repression, however this may be context dependant. For example, we see a modest enrichment for Cap-G binding at HP1 enriched chromatin, however, although thought to be a primarily repressive chromatin marker, some studies (e.g. Wit, Greil and van Steensel, 2007) show that HP1 is also associated with activate chromatin states. We would also like to point out that some of the gene expression changes may be the result of indirect effects, indeed since a large proportion of differentially expressed genes do not coincide with Cap-G binding sites, we presume that this is the case. Therefore, from these data we do not think it is that surprising that we do not see only de-repression of genes in our data.

It is possible that Cap-G acts in concert with the condensin complex to organise 3D chromatin structure in a manner similar to that reported for other SMC proteins in interphase. For example, cohesin is known to be involved in organising TAD boundaries. If acting in this kind of role, the consequences of Cap-G loss may be more difficult to predict than straightforward loss of gene repression.

With regards to the low numbers of overlapping differentially expressed genes between the elav and *nsyb* conditions we think that this is not entirely unsurprising when taking the binding data into consideration. We see that there is as around as much difference in binding of Cap-G between the two neuronal stages as we observe between either of the neurons and their progenitor cells (Figure 4—figure supplement 1C and Figure 4—figure supplement 2A). Therefore it is not unexpected to see that the loss of Cap-G in these cells has a different impact on transcriptomes. Also, this is consistent with the difference in phenotypes observed and the altered abundance of Cap-G seen in immunofluorescence experiments (Figure 1D).

3) The reviewers raised concerns that elav can be expressed in progenitors and thus felt that the authors cannot use the elav line as definitive evidence for postmitotic functions of CapG.They indicated that addition of a dividing cell marker (such as Dpn) in Figure 1—figure supplement 1A would be needed to validate that CapG is not reduced in dividing cells. Given that the authors show convincing behavioral data using the nsyb-KD line that indicates a function for Condensin in neurons, the conclusions based on the elav line could be toned down to admit possible expression in progenitors without undermining the story that the authors with to tell here.

The reviewers are right to be concerned about the possibility of premature knockdown of Cap-G in progenitor cells. We have provided multiple lines of evidence to show that this is unlikely to be a major contributing factor to the phenotypes observed in this case. To address the reviewers’ point, we have analysed further data to compare Cap-G levels between conditions in cells that are positively marked with the neuroblast marker Dpn. We see no significant difference in GFP levels between control and knockdown animals. These data along with representative images have been added to Figure 2—figure supplement 1B, C. Taking these data together, we believe it is unlikely that the phenotypes we have reported in elav-mediated knockdown animals are as a result of premature expression. However, we agree that premature degradation remains a possibility and are happy to change the text to reflect this fact. We have added a caveat to this effect in our Discussion (subsection “Differential requirement for Cap-G in newly born and mature neurons”).

With respect to the behavioral phenotypes in the nsyb line, although the reviewers were convinced that there were phenotypes (confirming functions of CapG in neurons) they did not think that detailed description of these phenotypes was highly meaningful given the survival deficits. Thus the commentary on CapG functions in behavior could also be written back.

We agree that these behavioural phenotypes should not be overinterpreted. We include these data only to provide a full characterisation of the neuronal dysfunction of these animals. It is important to note that since such a large proportion of flies do not make it to adulthood, these flies may have escaped the most severe consequences of Cap-G depletion. We have changed the final part of this section to reflect this in the text.